# MINI-CLUSTER GUIDED LONG-TAILED DEEP CLUSTERING

**Zhixin Li[1], Yuheng Jia[1,2,4]\*, Guanliang Chen[3], Hui Liu[4], Junhui Hou[5]\***

[1] School of Computer Science and Engineering, Southeast University, Nanjing 210096, China
[2] Key Laboratory of New Generation Artificial Intelligence Technology and Its
Interdisciplinary Applications (Southeast University), Ministry of Education, China
[3] Faculty of Information Technology, Monash University, Melbourne, 3800, Australia
[4] School of Computing Information Sciences, Saint Francis University, Hong Kong, China
[5] Department of Computer Science, City University of Hong Kong, Hong Kong, China
{lizhixin,yhjia}@seu.edu.cn, guanliang.chen@monash.edu,
h2liu@sfu.edu.hk, jh.hou@cityu.edu.hk

## ABSTRACT

As an important branch of unsupervised learning, deep clustering has seen substantial progress in recent years. However, the majority of current deep clustering methods operate under the assumption of balanced or near-balanced cluster distributions. This assumption contradicts the common long-tailed class distributions in real-world data, leading to severe performance degradation in deep clustering. Although many long-tailed learning methods have been proposed, these approaches typically rely on label information to differentiate treatment across different classes, which renders them inapplicable to deep clustering scenarios. How to re-weight the training of deep clustering models in an unsupervised setting remains an open challenge. To address this, we propose a mini-cluster guided long-tailed deep clustering method, termed MiniClustering. We introduce a specialized clustering head that divide data into much more clusters than the target number of clusters. These predicted clusters are referred to as mini-clusters. The mini-cluster-level predictions serve as the guide for estimating the appropriate weights for classes with varying degrees of long-tailedness. The weights are then incorporated to re-weight the self-training loss in model training. In this way, we can mitigate model bias by re-weighting gradients from different classes. We evaluate our method on multiple benchmark datasets with different imbalance ratios to demonstrate its effectiveness. Further, our method can be readily applied to the downstream of existing unsupervised representation learning frameworks for long-tailed deep clustering. It can also adapt label-dependent long-tailed learning methods to unsupervised clustering tasks by leveraging the estimated weights. The code is available at `https://github.com/LZX-001/MiniClustering`.

## 1 INTRODUCTION

Deep clustering is an unsupervised learning task, aiming to train deep neural networks to partition data points into clusters in the absence of labels. Existing deep clustering algorithms often assume a balanced class distribution in the dataset, which is inconsistent with the long-tailed distribution commonly seen in natural data (Li and Jia, 2025; Zhang et al., 2024a; Jiang et al., 2021). In real-world scenarios, data are often distributed in a long-tailed manner, where some classes (head classes) contain a large number of samples while others (tail classes) are represented by only a few. These long-tailed distributions pose a significant challenge in deep learning, as models tend to be biased toward head classes while underperforming on tail ones, leading to significant performance drop (Zhang et al., 2023). In supervised long-tailed learning, a common solution is to employ class-balancing strategies to mitigate such model bias: re-sampling the number of samples from different classes (Estabrooks et al., 2004; Liu et al., 2008; Zhang and Pfister, 2021), re-weighting each class

---

*Corresponding author: Yuheng Jia, Junhui Hou.

in the loss function (Lin et al., 2017; Cui et al., 2019; Ren et al., 2020), and adjusting the output logits of each class (Menon et al., 2021; Zhao et al., 2022; Tao et al., 2023). While these methods differ in form, their essence lies in assigning class-specific weights to head and tail classes, thereby differentially treat head and tail classes in training. To introduce such a re-balancing mechanism that mitigates model bias induced by data imbalance, knowing label frequencies as a necessary prior is typically required.

Unfortunately, deep clustering is totally unsupervised. Despite also facing severe performance degradation under long-tailed distributions (Li and Jia, 2025), it lacks labels and therefore cannot directly apply the aforementioned methods. To extend the principles of supervised long-tailed learning to long-tailed deep clustering, it is crucial to assess the appropriate weights for training each class in an unsupervised manner.

To address the aforementioned issues, we investigate the characteristics of clustering under long-tailed distributions and have found some important phenomena. We observe that head class samples often occupy a larger embedding space compared to tail class ones, which often leads to head class samples being assigned to multiple clusters while tail class samples end up sharing clusters with others. Therefore, we cluster the embeddings into a larger number of groups, referred to as mini-clusters, than the final target number of clusters, enabling finer-grained partitioning. In this way, average purity across clusters is improved. However, the number of mini-clusters is much larger than the desired number of clusters (referred to as target-clusters), which does not align with our objective for clustering. To establish the connection between target-clusters and mini-clusters, we analyze the sample assignments between them. We find that clusters representing head classes tend to be associated with a larger number of mini-clusters, while those corresponding to tail classes are linked to fewer. These phenomena are detailed in Section 3.1. Inspired by these phenomena, we introduce a network architecture with two clustering heads, where the target-cluster head outputs predictions over the desired number of clusters, and the mini-cluster head has a higher prediction dimensionality than target-clusters. Then, we decompose each target-cluster into its constituent mini-clusters. We align the predictions from the mini-cluster head and the target-cluster head to determine how many mini-clusters each target-cluster is divided into, thereby estimating the training weight for each class. In this manner, we enable the application of re-weighting strategies to long-tailed deep clustering in a way analogous to common supervised long-tailed learning method, effectively re-balancing the training process and alleviating model bias. We name this method MiniClustering, which is short for mini-cluster guided clustering.

In summary, the main contributions of this work are as follows.

• We reveal key phenomena in long-tailed clustering, based on which we propose an unsupervised method to estimate class weights, effectively re-weighting model training.

• We introduce a novel paradigm based on two clustering heads with different prediction granularities. Based on estimated class-specific weights, the proposed MiniClustering can effectively address long-tailed deep clustering. Extensive experiments showcase its superiority.

• We pioneer class-level re-weighting in unsupervised long-tailed learning, extending this strategy from supervised learning to deep clustering. MiniClustering enables supervised re-weighting losses to be adapted for

## 2 RELATED WORK

**Deep Clustering.** Deep clustering trains deep neural networks to assign unlabeled data points to distinct clusters, where the process is conducted without any manual annotations (Zhou et al., 2024). Compared to supervised learning, it can effectively save time and labor in label annotation. Deep clustering methods can be broadly categorized into two paradigms: those based on representation learning (Tao et al., 2020; Huang et al., 2023; 2014) and those based on clustering heads (Van Gansbeke et al., 2020; Li et al., 2021; Niu et al., 2022; Li et al., 2022; Qian, 2023). In contrast to traditional clustering (Hartigan and Wong, 1979; Von Luxburg, 2007; Xie et al., 2025; Zhang et al., 2025a;b), deep clustering based on representation learning employs neural networks to model and optimize data representations. They learn suitable embeddings through various approaches, such as autoencoders (Huang et al., 2014; Xie et al., 2016; Guo et al., 2017) and contrastive learning (Huang et al., 2023; Li et al., 2023; Tsai et al., 2021). Then clustering techniques such as K-means

(Hartigan and Wong, 1979) or spectral clustering (Von Luxburg, 2007) are applied to form cluster partitions. Deep clustering methods based on attached clustering heads typically introduce a clustering head following a feature extraction network and train it using pseudo-labels. Such approaches can directly produce clustering predictions without relying on additional techniques, making their architecture similar with the supervised learning paradigm (Van Gansbeke et al., 2020; Li et al., 2021; Qian, 2023; Tian et al., 2017; Jia et al., 2025; Li et al., 2025a). In general, most existing deep clustering methods are trained and evaluated primarily on balanced datasets, overlooking the long-tailed distributions commonly observed in natural data. As a result, their designs are often not robust to long-tailed data, leading to significant performance degradation (Li and Jia, 2025).

**Long-tailed Learning.** In reality, data distributions are often long-tailed, with head classes containing abundant samples and tail classes having scarce samples. Training models on such data can lead to biased models that are more prone to predicting common head classes while overlooking rare tail classes. When labels are involved during training, we can re-balance the model according to the frequency of each class's occurrence, thereby mitigating this bias. Common re-balancing techniques include: re-sampling balances the number of samples by over-sampling tail classes or under-sampling head classes (Chawla et al., 2002; Wang et al., 2019; Zhang and Pfister, 2021); re-weighting balances the class-wise contribution to gradient updates by assigning class-specific weights in the loss function according to label frequencies (Cui et al., 2019; Cao et al., 2019; Tan et al., 2020); logit adjustment balances the prediction difficulty across different classes by modifying the logits in biased models (Tian et al., 2020; Menon et al., 2021; Zhang et al., 2021b). These methods all require prior knowledge of labels, thereby applying varying degrees of influence to different classes. Consequently, they cannot be directly applied to unsupervised long-tailed learning. Methods for unsupervised long-tailed learning are scarce. Some approaches focus on unsupervised representation learning under long-tailed distributions (Jiang et al., 2021; Zhou et al., 2022; Li and Jia, 2025). They typically balance the loss between rare and frequent samples to mitigate model bias. However, they do not necessarily ensure the discriminability of the learned embeddings in long-tailed deep clustering. Other methods address long-tailed clustering as a downstream task by fine-tuning large pre-trained models (Zhang et al., 2024a;b). Due to the strong prior knowledge encoded in these models, their susceptibility to long-tailed bias is often less severe.

To the best of our knowledge, effectively training a clustering head for long-tailed deep clustering, following the principles of supervised long-tailed learning based on re-weighting, remains unexplored. Furthermore, how to accurately obtain class-specific weights to re-balance model training in an unsupervised manner is a critical yet under-investigated problem.

## 3 PROPOSED METHOD

### 3.1 MOTIVATION

Inspired by the aforementioned issues, we identify three important phenomena in long-tailed deep clustering.

**Phenomenon 1: Head classes occupy more embedding space than tail classes.** Following (Tang et al., 2020; Cao et al., 2019), we construct the long-tailed version of CIFAR-10 dataset (Krizhevsky et al., 2009) with an imbalance ratio of 10, where we conduct unsupervised representation learning using BYOL (Grill et al., 2020), SimCLR (Chen et al., 2020) and MoCo (He et al., 2020). These methods serve as the foundation for state-of-the-art deep clustering algorithms, as both representation-based approaches and those employing clustering heads typically rely on high-quality representation learning (Huang et al., 2023; Van Gansbeke et al., 2020; Li et al., 2021; Li and Jia, 2025). We record the confusion matrices of the K-means (Hartigan and Wong, 1979) clustering results based on the representations learned by these three methods, to illustrate the membership relationship between predicted clusters and ground-truth classes. In Figure 1, the class indices range from 0 to 9, with the number of samples decreasing progressively. In the confusion matrix, the intensity of each cell indicates the number of samples from a given ground-truth class (indexed by row) that are assigned to a specific predicted cluster (indexed by column). It can be observed that samples from head classes (i.e., classes with smaller indices) are predominantly distributed across multiple clusters (e.g., in SimCLR, samples from class 0 is primarily assigned to cluster 0, 7 and 8.). In contrast, the clusters to which tail classes are predominantly assigned may contain a large number of samples from other classes, and in some cases, these clusters are even domi-

nated by head classes (e.g., in SimCLR, class 9 and class 1 share the same cluster). K-means is a distance-based clustering method defined in Euclidean space, where each sample is assigned to the cluster whose centroid is nearest. This tends to produce clusters of comparable size. Under such a mechanism, head classes, due to their larger number of samples, naturally occupy a more extensive region in the embedding space and are thus partitioned into multiple clusters. Tail classes, which span a smaller region of the embedding space, are forced to share clusters with other classes.

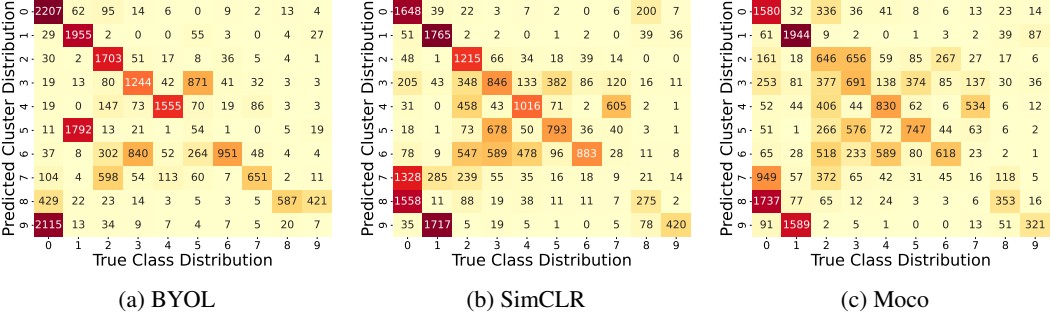

(a) BYOL  (b) SimCLR  (c) Moco

Figure 1: Confusion matrices of predicted cluster distributions and true class distributions for different methods. It can be observed that samples from head classes are often separated into multiple clusters, while samples from tail classes are forced to share a cluster with other categories. This indicates that, under long-tailed distributions, unsupervised training learns representations where head classes occupy a larger embedding space than tail classes.

**Phenomenon 2: Mini-clusters can enhance purity.** Directly utilizing current cluster information to guide training leads to inaccuracies, as some predicted clusters contain samples from multiple ground-truth classes while certain classes are distributed across multiple predicted clusters. A natural approach is to cluster the data into a larger number of clusters than the true number of classes, using a finer-grained division to increase the purity of each cluster, thereby facilitating subsequent processing. We perform clustering on the embeddings of CIFAR-10, which has 10 classes, using a higher number of clusters. We use purity, defined as the proportion of the largest class in a certain cluster, to measure these clusterings. As shown in Figure 2, this approach can improve the average purity across all clusters.

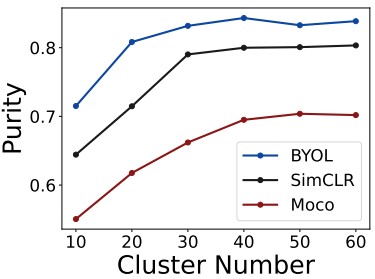

Figure 2: As the number of clusters increases, the average purity of all clusters is improved.

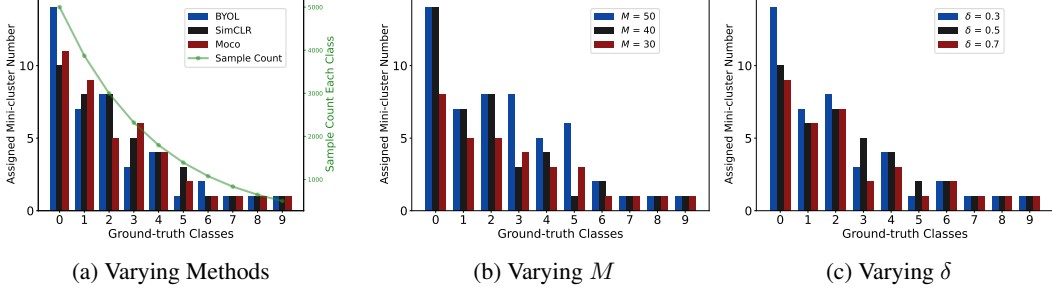

(a) Varying Methods  (b) Varying $M$  (c) Varying $\delta$

Figure 3: Across different methods, numbers of mini-clusters $M$, and the assignment threshold $\delta$, head classes are usually assigned a greater number of mini-clusters compared to tail ones.

**Phenomenon 3: Head classes are assigned more mini-clusters than tail classes.** We refer to the aforementioned fine-grained clusters as mini-clusters, while those with the number aligned to the desired or true number of classes are termed target-clusters. Inspired by the aforementioned

discussion, we aim to establish the relationship between the target-clusters and the mini-clusters. We conduct experiments on the long-tailed version of CIFAR-10 (Krizhevsky et al., 2009), where the number of samples in each class decreases exponentially from index 0 to 9. Assuming there are $M$ mini-clusters in total, we assign a mini-cluster to a target-cluster if the proportion its samples belonging to that target-cluster exceeds a predefined threshold $\delta$. Specifically, we evaluated the mini-cluster assignment for head and tail classes under varying methods, varying $M$ and varying $\delta$. We set the method to BYOL, $M = 40$ and $\delta = 0.3$ as the default configuration, and varied one variable at a time for analysis. As shown in Figure 3, across different methods, numbers of mini-clusters $M$, and the assignment threshold $\delta$, head classes are consistently assigned more mini-clusters than tail classes. This suggests that mini-cluster assignment can serve as an indicator of whether a class is a head or tail class, thus enabling re-balanced model training.

**Theoretical Analysis.** Previous observations have shown that head classes are often assigned more mini-clusters than tail classes. We now provide a rigorous theoretical derivation, proving that when mini-cluster purity and representation quality satisfy certain conditions, classes with more samples dominate a larger number of mini-clusters. We assume that samples from $K$ classes follow a long-tailed distribution and are clustered into $M$ mini-clusters. The purity of in each mini-cluster is at least $\rho$. Let $S_{\min}$ and $S_{\max}$ be the minimum and maximum numbers of samples in mini-clusters, respectively. A mini-cluster is said to be dominated by class $k$ if its purity with respect to class $k$ is at least $\rho$. Let $m_k$ denote the number of mini-clusters dominated by class $k$. Let $\epsilon_k$ denote the number of samples from class $k$ that lie outside the mini-clusters dominated by class $k$. Let $N_k$ denote the number of samples in class $k$. Then we can obtain below theorem:

**Theorem 1.** *For classes $i$ and $j$, if the number of samples $N_i > N_j$, then the number of occupied mini-clusters $m_i > m_j$ holds when*

$$\rho > \frac{N_j S_{\max}}{(N_i - \epsilon_i) S_{\min}}.$$

If $\rho$ is too small or $\epsilon_i$ is too large, it may lead to the failure of $m_i > m_j$ when $N_i > N_j$. Both the minimum purity $\rho$ of mini-clusters and the number $\epsilon_i$ of samples from class $i$ that lies outside the mini-clusters dominated by class $i$ depend on the quality of the embeddings. If the embedding quality is low and the model struggles to consistently group samples of the same class into the same mini-clusters, the above property may fail to hold. Therefore, in proposed MiniClustering, we use a model pre-trained with unsupervised representation learning to ensure a reasonable embedding distribution.

A detailed proof and analysis can be found in the Appendix L.

## 3.2 MINICLUSTERING

Inspired by previous observations, we propose a mini-cluster guided long-tailed deep clustering method, named MiniClustering. We introduce a clustering head training strategy based on the self-labeling architecture (Li et al., 2021; Van Gansbeke et al., 2020; Qian, 2023), which leverages mini-cluster-level predictions to balance the training losses of classes with varying long-tailed degrees.

The network architecture of the proposed method consists of three components: an encoder $f_e$ that extracts embeddings from the raw data; a target-cluster head $f_t$ that produces predictions with the same dimensionality as the predefined target number of clusters; a mini-cluster clustering head $f_m$ that outputs predictions at the finer-grained mini-cluster level. Both $f_t$ and $f_m$ are linear classifiers that take embeddings as input, but their output logit dimensions are the number of target-cluster $K$ and the number of mini-clusters $M$, respectively. This model is trained jointly with three losses, including mini-cluster self-training loss $\mathcal{L}_m$, re-weighted target-cluster self-training loss $\mathcal{L}_r$ and similarity alignment loss $\mathcal{L}_s$. The network architecture of MiniClustering is shown in Figure 4. After data augmentation, the transformed input data is fed into the encoder $f_e$ to obtain embeddings, which are then passed to the target-cluster head $f_t$ and the mini-cluster head $f_m$ respectively for prediction and a self-labeling strategy is applied for training unsupervisedly. The predictions from the mini-cluster head are matched with those from the target-cluster head according to a specific criterion described below. The number of mini-clusters belonging to each target-cluster determines the re-balancing weight for that class. Additionally, we align the similarities of outputs from the two clustering heads to prevent training desynchronization.

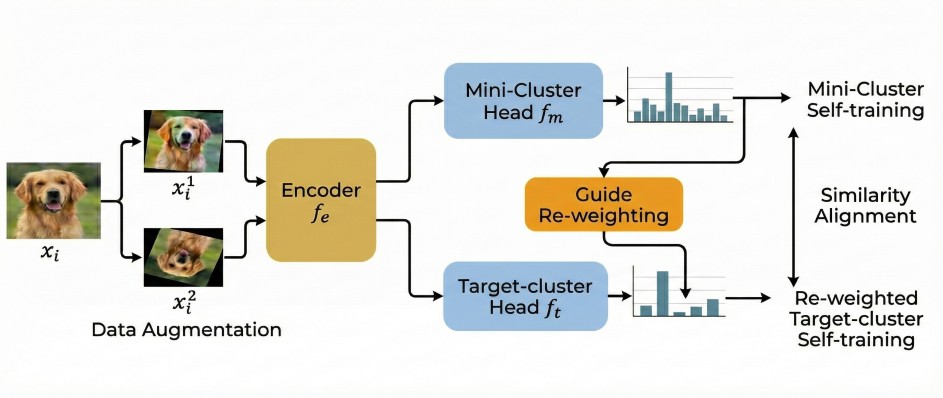

Figure 4: The network architecture of MiniClustering.

**Notations.** Given a batch $\mathcal{B}$ in the dataset, and $x_i \in \mathcal{B}$ is a sample in this batch, then $x_i^1$ and $x_i^2$ are two augmented views to serve as the input for encoder $f_e$. $f_e(x_i^1)$ and $f_e(x_i^2)$ are two embedding accordingly. They are both fed to target-cluster and mini-cluster head later. Specifically, $x_i^1$ is used for prediction, while $x_i^2$ is used to compute pseudo-labels. $p_i^t = \sigma(f_t(f_e(x_i^1))) \in \mathbb{R}^K$ and $p_i^m = \sigma(f_m(f_e(x_i^1))) \in \mathbb{R}^M$ are the the predictions from target-cluster head $f_t$ and mini-cluster head $f_m$, where $\sigma(\cdot)$ is the softmax function. $c_i^t = \max(p_i^t)$ and $c_i^m = \max(p_i^m)$ are confidences of the predictions respectively. $\hat{y}_i^t = \operatorname{argmax}(\sigma(f_t(f_e(x_i^2))))$ and $\hat{y}_i^m = \operatorname{argmax}(\sigma(f_t(f_e(x_i^2))))$ are the pseudo labels generated by the target-cluster head and mini-cluster head, respectively, for $x_i$. $y_i^t$ and $y_i^m$ denote one-hot vectors, where the positions corresponding to the pseudo labels are set to 1 respectively, and all other entries are 0. $|\cdot|$ denotes the size of the set. $(\cdot)^T$ is the transpose of the matrix.

**Mini-cluster self-training loss.** The mini-cluster head maps the input embeddings to a dimensionality higher than the number of target-clusters, enabling finer-grained predictions. To train the mini-cluster head, we adopt a threshold-based self-labeling strategy (Van Gansbeke et al., 2020) based on cross-entropy (CE) loss,

$$\mathcal{L}_m = -\frac{1}{|\mathcal{S}_\tau^m|} \sum_{i \in \mathcal{S}_\tau^m} \sum_{j=1}^M y_{i,j}^m \log(p_{i,j}^m), \quad \mathcal{S}_\tau^m = \{i \mid x_i \in \mathcal{B}, c_i^m > \tau\}, \tag{1}$$

where $\mathcal{S}_\tau^m$ is the set containing indices of samples whose confidences output from $f_m$ exceed the threshold $\tau$, $y_{i,j}^m$ and $p_{i,j}^m$ denote the $j_{\text{th}}$ element in the corresponding vectors. Through a threshold-based self-labeling method, we can obtain reliable pseudo-labels, thereby improving the quality of training.

**Re-weighted target-cluster self-training loss.** The predictions output by the target-cluster head $f_t$ correspond to the actual number of classes, but it faces an imbalanced class distribution. As illustrated in Section 3.1, head classes occupy a larger embedding space compared to tail classes. In the mini-cluster setup, samples from head classes are expected to be assigned to a greater number of high-purity mini-clusters, while those from tail classes are assigned to fewer. Consequently, the number of mini-clusters associated with each target-cluster can reflect the spatial distribution of the corresponding class. We posit that target-clusters occupying more mini-clusters are spatially more dispersed and are thus more likely to be head classes, whereas those occupying fewer mini-clusters are more likely to be tail classes. Inspired by common approaches in supervised long-tailed learning (Cui et al., 2019; Ren et al., 2020; Tan et al., 2020), we accordingly re-weight the loss for head and tail classes during model training to achieve re-balance and mitigate the model bias. The class-wise weights are estimated based on the number of mini-clusters occupied by each target-cluster. The weight for target-cluster $k$ is

$$w_k = \frac{M}{\max(\sum_{j=1}^M \mathbf{1}(\frac{|T_{k,j}|}{|T_j|} > \delta), 0.5)}, \tag{2}$$

where $T_{k,j} = \{i|x_i \in \mathcal{B}, \hat{y}_i^t = k, \hat{y}_i^m = j\}$ is the set containing indices of samples predicted to target-cluster $k$ and mini-cluster $j$ at the same time, $T_j = \{i|x_i \in \mathcal{B}, \hat{y}_i^m = j\}$ is the set containing indices of samples predicted to mini-cluster $j$. $\mathbf{1}(\cdot)$ is an indictor function, which returns 1 when the condition is satisfied else 0. The $\sum_{j=1}^{M} \mathbf{1}(\frac{|T_{k,j}|}{|T_j|} > \delta)$ in the Eq. (2) counts the number of mini-clusters occupied by samples from target-cluster $k$. We assign a mini-cluster to a target-cluster when over a threshold $\delta$ proportion of its samples are from that target. The weight of each target-cluster is then determined based on the number of mini-clusters affiliated with it. To prevent division by zero, we set a lower bound of 0.5 on the number of assigned mini-clusters, ensuring that target-clusters with no assigned mini-clusters have larger weights in training. Note that this is a soft assignment scheme, as theoretically, when the threshold $\delta$ is less than 0.5, the mini-cluster could be counted toward multiple target-clusters. In Eq. (2), larger numerators lead to smaller weights for head classes and, conversely, larger weights for tail classes. To re-balance gradient updates, we re-weight threshold-based self-labeling in below manner,

$$\mathcal{L}_r = -\frac{1}{|\mathcal{S}_\tau^t|} \sum_{i \in \mathcal{S}_\tau^t} \sum_{j=1}^{K} w_{\hat{y}_i^t} y_{i,j}^t \log(p_{i,j}^t), \quad \mathcal{S}_\tau^t = \left\{ i \,\middle|\, x_i \in \mathcal{B}, c_i^t > \tau \right\}, \tag{3}$$

where $\mathcal{S}_\tau^t$ is the set containing indices of samples whose confidences output from $f_m$ exceed the threshold $\tau$.

**Similarity alignment loss.** The target-cluster head and the mini-cluster head are updated by different losses, which can easily lead to misalignment during training, resulting in the failure of the re-weighting strategy and potential model training conflicts. Due to the different output dimensions of the target-cluster head and the mini-cluster head, we adopt a similarity-based alignment method to ensure consistency between the outputs of the two clustering heads. We consider $P^t$ and $P^m$ are matrices formed by row-wise stacking of the prediction results from $f_t$ and $f_m$ on the batch, respectively. They are of size $N \times K$ and $N \times M$, respectively, where $N$ is the batch size. As samples with close predictions in the target-clusters should also be close in the mini-clusters, the similarity matrices $P^t P^{t^T}$ and $P^m P^{m^T}$ should be close, too. We bring the similarity matrices of the two closer together using the mean squared error (MSE) loss,

$$\mathcal{L}_s = \frac{1}{N^2} \|P^t P^{t^T} - P^m P^{m^T}\|_F^2, \tag{4}$$

where $\| \cdot \|_F$ is the Frobenius norm. This loss is similar to the within/cross-graph consistency proposed in (Lin et al., 2023), which encourages the alignment of the edges across samples in similarity graphs.

**Overall procedure and objective.** Following (Cai et al., 2023), we initialize the network weights in $f_t$ and $f_m$ using the cluster centers obtained from K-means (Hartigan and Wong, 1979) to accelerate training. Additionally, we initialize the network bias to zero at the beginning. Please note that encoder $f_e$, target-cluster head $f_t$ and mini-cluster head $f_m$ are all in backpropagation. Before performing MiniClustering, the encoder $f_e$ can be pre-trained on long-tailed datasets in advance using any unsupervised representation learning and specific settings can be found in Section 4.1. The overall objective of MiniClustering is

$$\mathcal{L} = \mathcal{L}_r + \alpha \mathcal{L}_m + \beta \mathcal{L}_s, \tag{5}$$

where $\alpha$ and $\beta$ are trade-off parameters to balance three terms. After training, we use the predictions from the target-cluster head for clustering. The algorithmic pseudocode is summarized in Appendix B.

## 4 EXPERIMENTS

### 4.1 EXPERIMENT SETTINGS

**Datasets.** We conduct experiments on benchmark datasets, including CIFAR-10 (Krizhevsky et al., 2009), CIFAR-20 (Krizhevsky et al., 2009), STL-10 (Coates et al., 2011), Tiny ImageNet (Le and Yang, 2015) and ImageNet-LT (Liu et al., 2019). CIFAR-20 is CIFAR-100 using 20 super-classes as the cluster number. On STL-10, we use all data from both the training and test sets for training and clustering. Unless otherwise specified, for all other datasets, we use only the training set. Following

(Cao et al., 2019; Zhou et al., 2020), we use the imbalance ratio to determine the number of samples for each class in long-tail dataset construction. The number of samples per class follows an exponential decay, where the ratio between the largest and smallest classes is defined as the imbalance ratio.

**Implementation Details.** Following (Li and Jia, 2025), we adopt ResNet-18 as backbone for encoder $f_e$. Following (Van Gansbeke et al., 2020), linear classifiers are adopted as the clustering heads for $f_t$ and $f_m$. The number of target-clusters $K$, as in most deep clustering methods (Huang et al., 2023; Li et al., 2021; Van Gansbeke et al., 2020; Li and Jia, 2025; Qian, 2023; Hartigan and Wong, 1979), is a known prior, corresponding to the true number of classes. Mini-cluster number $M$ is a hyperparameter. For extracting meaningful embeddings, we adopt BYOL framework (Grill et al., 2020) and train $f_e$ in advance on the same long-tailed datasets as MiniClustering. We take the target network in BYOL as the encoder $f_e$. To align with the training epochs in (Li and Jia, 2025), we pre-train $f_e$ for 800 epochs and then execute MiniClustering for 200 epochs, unless otherwise specified. In MiniClustering, we adopt a batch size of 512, following (Li and Jia, 2025). We set a constant learning rate of 1e-5 and a self-labeling threshold $\tau$ of 0.99, unless otherwise specified. For other training details and hyperparameter settings of each experiment, please refer to Appendix C.

Table 1: Clustering performance in percentage (%) on three benchmark datasets (imbalance ratio = 5) across various methods.

| Datasets | CIFAR-10 | | | | CIFAR-20 | | | | STL-10 | | | |
|---|---|---|---|---|---|---|---|---|---|---|---|---|
| Metrics | ACC | CAA | NMI | ARI | ACC | CAA | NMI | ARI | ACC | CAA | NMI | ARI |
| SCAN | 55.6 | 60.3 | 54.0 | 39.3 | 38.8 | 39.7 | 42.6 | 25.1 | 48.8 | 49.8 | 46.2 | 34.9 |
| SDCLR | 44.1 | 50.5 | 43.4 | 38.0 | 40.2 | 38.7 | 40.3 | 23.8 | 35.8 | 37.4 | 34.4 | 19.3 |
| CC | 25.2 | 20.9 | 18.3 | 1.47 | 16.0 | 12.0 | 18.0 | 00.0 | 41.5 | 30.0 | 46.2 | 22.4 |
| IDFD | 56.7 | 63.0 | 51.8 | 37.1 | 31.2 | 30.7 | 30.5 | 16.2 | 41.6 | 41.2 | 39.6 | 24.3 |
| CoNR | 41.4 | 43.5 | 34.9 | 23.3 | 23.8 | 23.4 | 21.6 | 12.1 | 32.4 | 31.3 | 29.0 | 18.2 |
| ProPos | 51.4 | 59.2 | 52.2 | 34.1 | 40.7 | 39.3 | 42.6 | 26.4 | 37.2 | 39.3 | 34.5 | 21.8 |
| DMICC | 40.6 | 42.5 | 36.9 | 25.4 | 25.2 | 23.1 | 20.6 | 7.5 | 45.9 | 45.6 | 45.3 | 35.5 |
| SeCu | 56.1 | 59.0 | 54.5 | 40.3 | 34.8 | 33.7 | 35.8 | 19.9 | 51.6 | 50.8 | 48.2 | 39.1 |
| LFSS | 58.1 | 63.1 | 59.7 | 43.3 | 43.5 | 42.2 | 46.8 | 28.6 | 46.6 | 43.7 | 46.0 | 34.0 |
| ConMix | 61.6 | 65.4 | 59.8 | 45.6 | 42.8 | 41.0 | 43.9 | 27.7 | 48.9 | 49.7 | 48.8 | 35.9 |
| BYOL (Baseline) | 60.4 | 66.4 | 62.0 | 46.5 | 38.5 | 39.2 | 41.3 | 23.4 | 48.5 | 46.4 | 49.2 | 35.4 |
| MiniClustering (Ours) | **74.3** | **72.6** | **69.9** | **62.1** | **45.6** | **42.3** | **47.5** | **30.7** | **54.7** | **52.7** | **54.4** | **42.8** |

Table 2: Clustering performance in percentage (%) on three benchmark datasets (imbalance ratio = 10) across various methods.

| Datasets | CIFAR-10 | | | | CIFAR-20 | | | | STL-10 | | | |
|---|---|---|---|---|---|---|---|---|---|---|---|---|
| Metrics | ACC | CAA | NMI | ARI | ACC | CAA | NMI | ARI | ACC | CAA | NMI | ARI |
| SCAN | 52.0 | 50.1 | 59.4 | 43.8 | 37.0 | 32.4 | 37.8 | 23.1 | 47.1 | 45.4 | 47.5 | 34.2 |
| SDCLR | 38.9 | 44.3 | 42.5 | 26.5 | 37.8 | 35.9 | 39.6 | 22.9 | 34.6 | 37.9 | 32.2 | 17.3 |
| CC | 40.6 | 27.5 | 43.9 | 18.8 | 19.9 | 14.3 | 21.9 | 1.1 | 43.0 | 35.3 | 44.7 | 25.4 |
| IDFD | 47.5 | 54.9 | 48.4 | 33.1 | 28.7 | 27.2 | 28.6 | 15.1 | 38.6 | 34.7 | 36.8 | 22.8 |
| CoNR | 31.4 | 44.3 | 29.2 | 17.8 | 20.3 | 17.9 | 17.3 | 8.0 | 34.8 | 32.5 | 30.7 | 21.0 |
| ProPos | 46.1 | 49.3 | 52.5 | 34.2 | 36.8 | 33.6 | 40.1 | 22.5 | 35.6 | 38.2 | 37.2 | 23.9 |
| DMICC | 36.6 | 39.5 | 36.8 | 25.9 | 24.7 | 21.9 | 20.7 | 10.1 | 41.3 | 41.9 | 38.7 | 30.3 |
| SeCu | 51.5 | 57.0 | 54.8 | 39.6 | 32.0 | 30.7 | 33.3 | 17.6 | 48.2 | 49.6 | 50.0 | 36.8 |
| LFSS | 56.3 | 59.7 | 57.9 | 43.0 | 41.1 | 40.0 | 46.8 | 28.4 | 43.3 | 42.3 | 45.4 | 30.8 |
| ConMix | 53.3 | 58.2 | 57.1 | 40.8 | 41.7 | 39.3 | 43.6 | 27.0 | 47.4 | 48.7 | 48.2 | 33.9 |
| BYOL (Baseline) | 51.9 | 55.2 | 56.3 | 41.7 | 39.8 | 37.1 | 41.9 | 25.0 | 41.2 | 41.8 | 46.8 | 29.6 |
| MiniClustering (Ours) | **64.6** | **61.4** | **63.9** | **56.7** | **44.3** | **41.5** | **47.2** | **30.4** | **52.0** | **54.4** | **50.8** | **39.1** |

## 4.2 MAIN RESULTS

Following (Li and Jia, 2025), we conduct experiments on CIFAR-10, CIFAR-20 and STL-10 with imbalance ratios of 5 and 10 to evaluate clustering performance of MiniClustering. We compare state-of-the-art methods, including SCAN (Van Gansbeke et al., 2020), SDCLR (Jiang et al., 2021), CC (Li et al., 2021), IDFD (Tao et al., 2020), CoNR (Yu et al., 2023), ProPos (Huang et al., 2023), DMICC (Li et al., 2023), SeCu (Qian, 2023), LFSS (Li et al., 2025b) and ConMix (Li and Jia, 2025). We adopt BYOL (Grill et al., 2020), which is used for pre-training the encoder $f_e$, as the baseline, and apply K-means (Hartigan and Wong, 1979) to obtain its cluster assignments. We reproduced SCAN, SeCu, and LFSS on the long-tailed datasets following their original experimental configurations. Results of the other methods are cited from ConMix (Li and Jia, 2025). Following (Li and

Jia, 2025), we assess the effectiveness of long-tailed deep clustering across four metrics: accuracy (ACC), class-averaged accuracy (CAA), normalized mutual information (NMI), and adjusted rand index (ARI). The comparison results of our method against various approaches are presented in Table 1 and 2, with the best performance highlighted in bold.

It can be observed that MiniClustering achieves a significant advantage over other state-of-the-art approaches. It achieves the best performance across all metrics under all experimental settings. Compared to the baseline BYOL, the proposed approach shows particularly significant improvement. For example, on CIFAR-10 with imbalance ratios of 5 and 10, MiniClustering outperforms BYOL by 13.9% and 12.7% in ACC, and achieves improvements of 15.6% and 15.0% in ARI, respectively. The experimental results demonstrate the effectiveness of MiniClustering in addressing the long-tailed deep clustering problem.

To comprehensively evaluate the effectiveness of our method, we conducted extensive experiments, including ablation studies (detailed in Section 4.3), hyperparameter analysis (detailed in Appendix D), evaluation of incorporating MiniClustering into downstream of unsupervised representation learning framework (detailed in Section 4.4), applying supervised long-tailed learning losses into MiniClustering (detailed in Section 4.5), facing extreme class imbalance (detailed in Appendix E ), validation on large-scale datasets (detailed in Appendix F), statistical significance analysis (detailed in Appendix H), visualization (detailed in Appendix I), evaluation on text datasets (detailed in Appendix J). These results collectively demonstrate the robustness and practical utility of our approach.

## 4.3 ABLATION STUDY

The ablation studies are shown in Table 3. Compared to the baseline BYOL, we achieve significant performance improvements, demonstrating the effectiveness of the MiniClustering strategy as a whole. If we only use threshold-based self-labeling to train a single target-cluster-level clustering head, performance improves over the baseline but remains far inferior to MiniClustering. This not only validates the rationality of our use of self-labeling scheme, but also

Table 3: Ablation studies on CIFAR-10 with an imbalance ratio of 10.

|  | ACC | CAA | NMI | ARI |
| --- | --- | --- | --- | --- |
| BYOL | 51.9 | 55.2 | 56.3 | 41.7 |
| Only threshold-based self-labeling | 55.6 | 57.6 | 66.3 | 52.5 |
| MiniClustering w/o threshold $\tau$ constraint | 54.1 | 55.1 | 58.5 | 45.6 |
| Only $\mathcal{L}_r$ | 59.9 | 58.1 | 63.5 | 52.0 |
| MiniClustering w/o $\mathcal{L}_r$ | 46.7 | 39.0 | 51.1 | 26.7 |
| MiniClustering w/o $\mathcal{L}_m$ | 64.5 | 43.6 | 65.4 | 50.3 |
| MiniClustering w/o $\mathcal{L}_s$ | 56.0 | 57.6 | 63.5 | 40.0 |
| MiniClustering | 64.6 | 61.4 | 63.9 | 56.7 |

demonstrates the effectiveness of the proposed mini-cluster guided re-weighting strategy. It is worth noting that self-labeling training achieves a higher NMI score than MiniClustering, but significantly lower ACC and CAA. This suggests that standard self-labeling tends to merge samples from tail classes into other clusters. The same phenomenon occurs when $\mathcal{L}_m$ is absent, indicating that $\mathcal{L}_m$ helps separate tail classes from other classes during prediction.

We can also see that without a threshold to limit the samples involved in loss calculation in MiniClustering, the performance would drop. We tested training with only the $\mathcal{L}_r$ involved. In this scenario, the loss of the target-cluster head is effectively re-weighted, but it is not synchronized with the mini-cluster head, leading to a performance drop. When $\mathcal{L}_r$ is absent in training, the weight of target-cluster head $f_t$ is not updated, leading to poor performance. When $\mathcal{L}_s$ is absent, the two clustering heads may desynchronize, reducing performance.

## 4.4 MINICLUSTERING AS A DOWNSTREAM COMPONENT IN UNSUPERVISED REPRESENTATION LEARNING PIPELINES

MiniClustering operates on an encoder $f_e$ pre-trained via representation learning to generate suitable embeddings, and can be readily applied downstream of various unsupervised representation learning frameworks. While we report performance using BYOL (Grill et al., 2020) as the backbone, the method demonstrates consistent effectiveness across other frameworks as well. We conduct experiments on CIFAR-10 and STL-10 with an imbalance ratio of 10, employing encoders pre-trained with SimCLR and MoCo. The results in Table 4 demonstrate that MiniClustering keeps effective

when applied with different pre-training frameworks, confirming its compatibility and robustness across representation learning methods.

Table 4: Clustering performance in percentage (%) on CIFAR-10 and STL-10 (imbalance ratio = 10) based on various frameworks.

| Datasets | | CIFAR-10 | | | | STL-10 | | | |
|---|---|---|---|---|---|---|---|---|---|
| Frameworks | Methods | ACC | CAA | NMI | ARI | ACC | CAA | NMI | ARI |
| BYOL | Baseline | 51.9 | 55.2 | 56.3 | 41.7 | 41.2 | 41.8 | 46.8 | 29.6 |
| | MiniClustering | **64.6** | **61.4** | **63.9** | **56.7** | **47.7** | **54.3** | **48.1** | **32.5** |
| SimCLR | Baseline | 43.4 | 47.9 | 45.8 | 30.4 | 38.1 | 38.8 | 37.5 | 21.6 |
| | MiniClustering | **51.5** | **58.5** | **55.5** | **37.5** | **41.4** | **41.1** | **40.3** | **24.9** |
| Moco | Baseline | 37.9 | 41.1 | 38.1 | 26.3 | 30.5 | 30.6 | 21.1 | 12.8 |
| | MiniClustering | **41.7** | **47.0** | **44.3** | **29.6** | **35.4** | **39.8** | **30.4** | **19.1** |

## 4.5 APPLYING EXISTING LONG-TAILED LEARNING LOSSES IN MINICLUSTERING

In MiniClustering, we estimate the weight for each target-cluster based on the number of mini-clusters assigned to it, and directly apply re-weighting to the softmax loss. In supervised long-tailed learning, studies have proposed various re-weighting schemes based on class weights (Zhang et al., 2023). Although these methods originally require label frequencies to compute the weights, our proposed approach enables their adaptation to deep clustering by providing an unsupervised estimation of class-specific weights, thereby extending their applicability to the unsupervised setting. We select several well-established and empirically effective re-weighting methods, including CB Softmax (Cui et al., 2019), BALMS (Ren et al., 2020) and EQL (Tan et al., 2020), to validate the generality and robustness of the weights estimated by MiniClustering. We replace $\mathcal{L}_r$ with the respective loss functions from these works and employ the weights estimated by our method, conducting experiments on CIFAR-10 and STL-10 with an imbalance ratio of 10. As shown in Table 5, our approach successfully extends existing supervised long-tailed learning losses to the unsupervised clustering setting, achieving competitive performance. In some cases their performance even surpass the original MiniClustering, because MiniClustering uses only the basic weighted softmax loss for demonstration. The details of these losses are in Appendix G.

Table 5: Clustering performance of employing various long-tailed learning losses in our scheme on CIFAR-10 and STL-10 (imbalance ratio = 10).

| Datasets | CIFAR-10 | | | | STL-10 | | | |
|---|---|---|---|---|---|---|---|---|
| Losses | ACC | CAA | NMI | ARI | ACC | CAA | NMI | ARI |
| Ours | 64.6 | 61.4 | 63.9 | 56.7 | **52.0** | **54.4** | **50.8** | **39.1** |
| CB Softmax | 64.1 | 61.3 | 63.0 | 55.3 | 49.1 | 53.3 | 47.8 | 33.6 |
| BALMS | 67.3 | **61.9** | 64.5 | 60.0 | 47.8 | 52.9 | 47.3 | 33.1 |
| EQL | **67.4** | **61.9** | **64.7** | **60.2** | 46.8 | 53.0 | 47.7 | 31.9 |

## 5 CONCLUSION

In this paper, we proposed a mini-cluster guided long-tailed deep clustering method MiniClustering, aiming to address the critical problem of how to re-weight model training in an unsupervised manner. We introduced an additional mini-cluster head to enable fine-grained predictions and enhance the average cluster purity. By aligning the predictions from the target-cluster head with those from the mini-cluster head, we determine the appropriate weight for each target-cluster. We demonstrated the effectiveness of our approach across multiple datasets and further show that it can be seamlessly integrated into various unsupervised representation learning frameworks. Moreover, the estimated weights can be applied to different long-tailed learning losses, effectively extending previously proposed supervised long-tailed learning methods to the domain of long-tailed deep clustering.

## ACKNOWLEDGMENTS

This work was supported in part by the National Natural Science Foundation of China under Grants U24A20322, 62576094 and 62422118, in part by the Hong Kong UGC under grants UGC/FDS11/E03/24, UGC/FDS11/E03/25, and in part by the Hong Kong Research Grants Council under Grants 11219324 and N_CityU1114/25. This research work was also supported by the Big Data Computing Center of Southeast University.

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

## A    STATEMENT ON THE USE OF LARGE LANGUAGE MODELS (LLMS)

We used the LLM solely to assist with writing, including grammar and spell checking, and improving sentence clarity to enhance readability. All content generated by the LLM was manually checked by the authors, who take full responsibility for it. The LLM was not involved in any other aspects of the paper. The writing logic of every sentence, as well as all experimental designs and conceptual contributions, were developed and completed by the authors.

## B  ALGORITHMIC PSEUDOCODE

To improve readability, the complete workflow of the MiniClustering is illustrated in the following algorithm:

---

**Algorithm 1** MiniClustering

---

**Require:** Long-tailed dataset $\mathcal{D}$, pre-trained encoder $f_e$, Number of target-clusters $K$ and mini-cluster $M$, self-training threshold $\tau$, assignment threshold $\delta$, trade-off parameters $\alpha$ and $\beta$, epoch number $T$.

**Ensure:** $K$ target-clusters.
1: Initialize target-cluster head $f_t$ and mini-cluster head $f_m$ using K-means.
2: **for** $t = 1$ to $T$ **do**
3:      Sample a batch $\{x_i\}_{i=1}^{N}$
4:      Compute the prediction $\{p_i^m\}_{i=1}^{N}$, confidence score $\{c_i^m\}_{i=1}^{N}$, and pseudo-label $\{\hat{y}_i^m\}_{i=1}^{N}$ from the mini-cluster head.
5:      Compute mini-cluster self-training loss $\mathcal{L}_m$ by Eq. (1).
6:      Compute the prediction $\{p_i^t\}_{i=1}^{N}$, confidence score $\{c_i^t\}_{i=1}^{N}$, and pseudo-label $\{\hat{y}_i^t\}_{i=1}^{N}$ from the target-cluster head.
7:      Compute class-specific weights $\{w_k\}_{k=1}^{K}$ by Eq. (2).
8:      Compute re-weighted target-cluster self-training loss $\mathcal{L}_r$ by Eq. (3).
9:      Compute similarity alignment loss $\mathcal{L}_s$ by Eq. (4).
10:      Compute overall objective $\mathcal{L}$ by Eq. (5).
11:      Update $f_e$, $f_t$ and $f_m$.
12: **end for**
13: Predict on the data in $\mathcal{D}$ using $f_t$.

---

## C  TRAINING DETAILS AND HYPERPARAMETER SETTINGS

We provide additional experimental details here that are not mentioned in the main text due to space constraints. Following (Chen et al., 2020; Li and Jia, 2025), for all experiments on CIFAR-10 and CIFAR-20, we set the first convolution layer to have a kernel size of 3×3 and a stride of 1, and remove the first max-pooling layer in ResNet-18 due to relatively small image sizes. We adopt the data augmentation in (Chen et al., 2020). There are four hyperparameters in MiniClustering, including mini-cluster assignment threshold $\delta$, mini-cluster number $M$, and two trade-off parameters $\alpha$ and $\beta$. To facilitate a better reproduction of our method, we provide the default hyperparameter settings used to obtain the reported results on different datasets. On CIFAR-10 (imbalance ratio = 5), we set mini-cluster number $M$ to 30, the assignment threshold $\delta$ to 0.7, trade-off parameters $\alpha$ and $\beta$ to 0.5 and 1. On CIFAR-10 (imbalance ratio = 10), we set $M$ to 40, $\delta$ to 0.3, $\alpha$ to 0.2, and $\beta$ to 0.5. On CIFAR-20 (imbalance ratio = 5), we set $M$ to 30, $\delta$ to 0.3, $\alpha$ to 0.5, and $\beta$ to 0.8. On CIFAR-20 (imbalance ratio = 10), we set $M$ to 30, $\delta$ to 0.3, $\alpha$ to 0.2, and $\beta$ to 1. On STL-10 (imbalance ratio = 5), we set $M$ to 20, $\delta$ to 0.7, $\alpha$ to 1, and $\beta$ to 0.2. On STL-10 (imbalance ratio = 10), we set $M$ to 40, $\delta$ to 0.3, $\alpha$ to 1, and $\beta$ to 0.2. On CIFAR-10 (imbalance ratio = 100), we set $M$ to 20, $\delta$ to 0.3, $\alpha$ to 1, and $\beta$ to 1.

## D  HYPERPARAMETER ANALYSIS

We present a detailed hyperparameter analysis using the control variates method. The effect of a single hyperparameter is examined while keeping all others fixed at their default values (please refer to Appendix C). The results are shown in Figure 5. We conduct experiments on CIFAR-10 and STL-10 with an imbalance ratio of 10, using various hyperparameter settings. Our method consistently outperforms the baseline across the vast majority of hyperparameter combinations, except in one particular hyperparameter configuration. This demonstrates the effectiveness of our method under different hyperparameter settings, i.e., CAA is slightly lower than that of the baseline when $M = 50$ on CIFAR-10. This demonstrates the effectiveness of our method under different hyperparameter settings. The clustering performance varies within a reasonable range when different

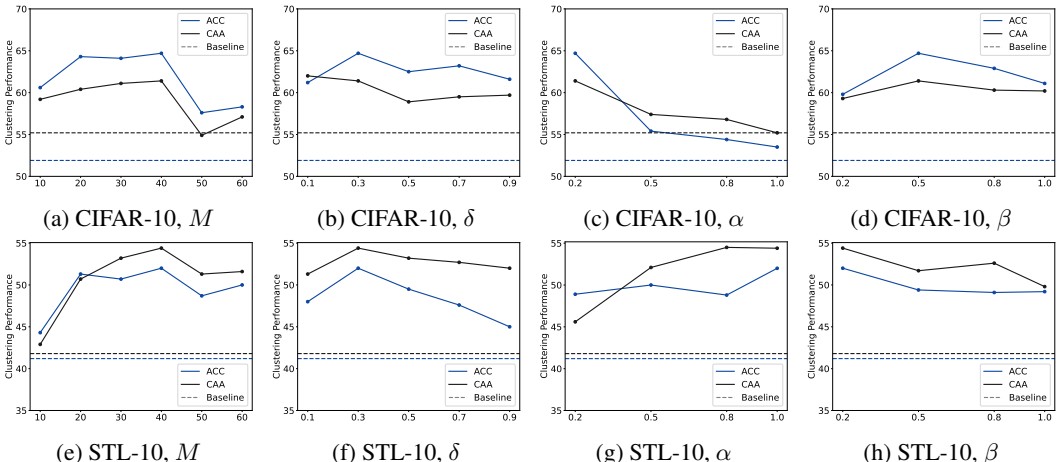

Figure 5: Hyperparameter analysis on CIFAR-10 and STL-10 with an imbalance ratio of 10.

hyperparameters are chosen. To achieve better performance, the following recommendations are worth considering:

**The number of mini-clusters $M$.** We suggest selecting a value greater than the actual number of classes but no more than four times that number. When $M$ is set equal to the actual number of classes, the resulting partitioning closely resembles that of the target clusters, thereby failing to leverage the benefits offered by mini-clusters for performance improvement. For instance, on STL-10, setting $M$=10 leads to a noticeable performance drop. Conversely, when M is too large, the overly fine-grained partitioning hinders the encoder from learning representations well-suited for predicting target clusters. For example, on CIFAR-10, performance degrades when $M$ is set to 50 or 60.

**Mini-cluster assignment threshold $\delta$.** We find it relatively stable across various settings and recommend using 0.3 or 0.5, which assigns more mini-clusters to each target cluster. And a more inclusive choice of $\delta$ is also beneficial when embedding quality is poor.

**Trade-off parameters $\alpha$ and $\beta$.** $\alpha$ has a more pronounced impact on performance. For instance, CIFAR-10 tends to favor smaller values of $\alpha$, while STL-10 prefers larger values. This behavior may be related to the intrinsic characteristics of the datasets. Therefore, we suggest using 0.5 as a default setting for $\alpha$, if no prior knowledge about the dataset is available. In contrast, the choice of $\beta$ exhibits relatively stable performance across different values, with most settings yielding comparable results.

# E MINICLUSTERING UNDER EXTREME IMBALANCE

To validate MiniClustering under extreme imbalance, we follow ConMix (Li and Jia, 2025) and evaluate on CIFAR-10 with an imbalance ratio of 100. As shown in Table 6, our method maintains superior performance, demonstrating its effectiveness and generalizability in long-tailed scenarios.

Table 6: Clustering performance in percentage (%) on CIFAR-10 (imbalance ratio = 100) across various methods.

| Methods | SimCLR | SDCLR | BYOL | SCAN | CoNR | ProPos | DMICC | SeCu | LFSS | ConMix | Ours |
|---|---|---|---|---|---|---|---|---|---|---|---|
| ACC | 32.2 | 32.8 | 33.5 | 35.4 | 28.1 | 38.2 | 31.3 | 25.1 | 37.7 | 40.4 | **46.9** |
| CAA | 31.6 | 32.3 | 35.5 | 40.4 | 20.9 | 38.3 | 26.1 | 22.2 | 28.5 | 38.5 | **44.6** |
| NMI | 36.4 | 39.4 | 43.1 | 48.1 | 25.6 | 42.4 | 41.1 | 15.0 | 46.1 | 53.4 | **54.0** |
| ARI | 20.2 | 22.2 | 25.3 | 28.2 | 14.6 | 24.9 | 22.4 | 10.6 | 28.4 | 33.5 | **34.6** |

# F MAXICLUSTERING ON LARGE-SCALE DATASETS

Since the mini-cluster head of our method needs to generate predictions for a number of clusters several times greater than target K, it may impose substantial training overhead on datasets with a very large number of target clusters. To validate the generalizability of our method on large-scale datasets with a high number of classes, we propose a conceptually analogous variant termed MaxiClustering. In contrast to the mini-cluster head, which produces a number of clusters exceeding the ground-truth class count, the maxi-cluster head generates a reduced number of clusters. In the network architecture, we replace the mini-cluster head with a maxi-cluster head, while keeping all other components unchanged. We propose the below weight estimation strategy for MaxiClustering:

$$w_k = \max(\sum_{j=1}^{M} \mathbf{1}(\frac{|T_{k,j}|}{|T_k|} > \delta), 0.5), \qquad (6)$$

where $T_{k,j} = \{i|x_i \in \mathcal{B}, \hat{y}_i^t = k, \hat{y}_i^m = j\}$ is the set containing indices of samples predicted to target-cluster $k$ and maxi-cluster $j$ at the same time, $T_k = \{i|x_i \in \mathcal{B}, \hat{y}_i^t = k\}$ is the set containing indices of samples predicted to target-cluster $k$. MaxiClustering and MiniClustering share the same loss functions, differing only in their weight estimation strategies.

In the MaxiClustering setting, most target-clusters are typically assigned to only one maxi-cluster due to the high similarity among samples from the same class. However, some target-clusters are assigned to multiple maxi-clusters, which arises from inconsistency between the partitions of maxi-clusters and target-clusters. Under such circumstances, because target-clusters corresponding to tail classes contain fewer samples, they are more likely to be assigned multiple maxi-clusters under the same assignment threshold. And they should be assigned higher weights during training to compensate for their scarcity.

To evaluate the effectiveness of MaxiClustering, we conduct experiments on Tiny ImageNet (Le and Yang, 2015) with an imbalance ratio of 10 and ImageNet-LT (Liu et al., 2019), following Li and Jia (2025). Tiny ImageNet contains 200 classes. To be consistent with ConMix (Li and Jia, 2025) in experimental setting, we only use the training set for constructing the long-tailed version. Also, we adopt the same experimental setup for MaxiClustering as used in the main results. We reproduce SCAN (Van Gansbeke et al., 2020), SeCu (Qian, 2023), and LFSS (Li et al., 2025b), while results for the other methods were quoted from ConMix (Li and Jia, 2025). The clustering results in Table 7 demonstrate the effectiveness of this approach in large-scale, long-tailed clustering scenarios.

Table 7: Clustering performance in percentage (%) on Tiny ImageNet (imbalance ratio = 10) across various methods.

| Methods | SimCLR | SDCLR | BYOL | SCAN | CoNR | ProPos | DMICC | SeCu | LFSS | ConMix | Ours |
|---------|--------|-------|------|------|------|--------|-------|------|------|--------|------|
| ACC | 16.0 | 14.6 | 16.1 | 15.9 | 8.8 | 15.0 | 9.27 | 11.5 | 22.9 | 16.5 | **25.5** |
| CAA | 14.2 | 12.9 | 14.0 | 15.7 | 8.2 | 13.2 | 8.8 | 8.7 | 18.4 | 14.5 | **19.4** |
| NMI | 33.9 | 35.2 | 34.2 | 38.8 | 28.2 | 34.4 | 28.0 | 39.7 | 28.7 | 34.9 | **40.6** |
| ARI | 7.9 | 8.6 | 8.6 | 11.8 | 3.9 | 7.5 | 3.7 | 8.5 | 14.0 | 8.3 | **14.9** |

ImageNet-LT is a long-tailed subset of ImageNet-1K (Deng et al., 2009), comprising 115.8K images across 1,000 classes, with the number of samples per class ranging from 5 to 1280. We following the setting in (Li and Jia, 2025), training a ResNet-50 using BYOL (Grill et al., 2020) for 160 epochs and then perform MaxiClustering for 40 epochs. Performance of other compared methods are cited from (Li and Jia, 2025). As shown in Table 8, MaxiClustering outperforms other methods. To facilitate the reproduction of our results, we provide a detailed description of the experimental

Table 8: Clustering performance in percentage on (%) ImageNet-LT

| Methods | SimCLR | SDCLR | BYOL | IDFD | CoNR | DMICC | ConMix | Ours |
|---------|--------|-------|------|------|------|-------|--------|------|
| ACC | 14.7 | 13.7 | 14.8 | 4.42 | 6.38 | 5.24 | 15.4 | **15.7** |
| CAA | 11.3 | 10.5 | 10.9 | 5.58 | 6.85 | 5.94 | 12.2 | **12.8** |
| NMI | 51.4 | 50.3 | 50.6 | 35.6 | 38.9 | 37.9 | 51.6 | **52.3** |
| ARI | 9.14 | 9.30 | 10.2 | 1.17 | 2.19 | 1.83 | 11.4 | **11.7** |

setup. For MaxiClustering on Tiny ImageNet (imbalance ratio = 10), we set constant learning rate to 1e-6, $\tau$ to 0.9, $M$ to 100, $\delta$ to 0.3, $\alpha$ to 1 and $\beta$ to 0.5. The training consists of 800 epochs of BYOL, followed by 200 epochs of MaxiClustering, consistent with the experimental settings in the main results. For MaxiClustering on ImageNet-LT, we set constant learning rate to 1e-5, $\tau$ to 0.5, $M$ to 200, $\delta$ to 0.3, $\alpha$ to 1 and $\beta$ to 1. The training consists of 160 epochs of BYOL, followed by 40 epochs of MaxiClustering, where we use ResNet-50 as the backbone for encoder $f_e$. This setting differs from our common experimental configuration.

## G  LONG-TAILED LOSSES IN DETAIL

In Section 4.5, we demonstrate that existing loss functions for supervised long-tailed learning (Cui et al., 2019; Ren et al., 2020; Tan et al., 2020) can be integrated with MiniClustering. Our specific approach is to replace $\mathcal{L}_r$ with the corresponding long-tailed learning losses, and substitute the weights calculated based on label frequencies in the original method with our estimated weights, leaving everything else unchanged in MiniClustering. Here we present the formulas for these long-tailed losses.

CB Softmax (Cui et al., 2019) uses the number of samples per class as weights to calculate the effective number for each class for re-weighting softmax loss. Similarly, we leverage the number of mini-clusters assigned to each target-cluster, i.e., $M/w_{\hat{y}_i^t}$ in Eq. (2) accordingly.

$$\mathcal{L}_{\text{CB-Softmax}} = -\frac{1}{|\mathcal{S}_\tau^t|} \sum_{i \in \mathcal{S}_\tau^t} \sum_{j=1}^K \frac{1-\gamma}{1-\gamma^{M/w_{\hat{y}_i^t}}} y_{i,j}^t \log(p_{i,j}^t), \quad \mathcal{S}_\tau^t = \left\{ i \,\middle|\, x_i \in \mathcal{B}, c_i^t > \tau \right\}, \quad (7)$$

where $\gamma$ is a hyperparameter. The weight in Eq. (1) is replaced by the new weight derived from the effective number based method, serving the same purpose to reduce the loss contribution from head classes and increase it for tail classes.

BALMS (Ren et al., 2020) re-weights softmax function according to label frequencies. We adopt this re-weight softmax loss in Eq. (3) as follow,

$$\sigma_{\text{BALMS}}(z_i) = \frac{\pi_i \exp(z_i)}{\sum_j \pi_j \exp(z_j)}, \quad (8)$$

where $z_i$ is the $i_{th}$ element of $z$, indicating the logit for the $i_{th}$ target-cluster. $\pi_i$ represents the proportion of mini-clusters assigned to the $i_{th}$ target-cluster relative to the total. We replace the softmax function in $\mathcal{L}_r$ with this one and cancel the weight outside the softmax loss.

EQL (Tan et al., 2020) re-weights softmax loss as well,

$$\sigma_{\text{EQL}}(z_i) = \frac{\exp(z_i)}{\sum_j w_j \exp(z_j)}, \quad (9)$$

where $w_j$ represents the weights estimated in Eq (2) for $j_{th}$ target-cluster. In this way, we allow tail classes to have larger gradients while head classes have smaller gradients.

## H  STATISTICAL SIGNIFICANCE ANALYSIS

To demonstrate that the performance improvement is not due to random chance, we ran our method multiple times using different random seeds and computed the mean and standard deviations of its performance. For a fair comparison, we reproduced the most competitive methods, including SeCu, LFSS, and ConMix, for evaluation. Experiments were conducted on imbalanced versions of CIFAR-10 and CIFAR-20 with an imbalance ratio of 10. We repeated each experiment five times using the same random seeds for all methods. The results are presented in Table 9. As shown, our method consistently achieves significantly better performance in terms of mean values compared to other approaches, especially exhibiting a clear advantage on the CIFAR-10 dataset. We use the t-test for significance testing and mark with an asterisk (*) the metrics where our method is significantly better than other methods at the 95% confidence level. Our method is significantly better than the most competitive baseline in the majority of cases.

Table 9: Statistical significance analysis of clustering performance on CIFAR-10 and CIFAR-20 (imbalance ratio = 10), with each method run five times using different random seeds.

| Method | CIFAR-10 | | | | CIFAR-20 | | | |
|---|---|---|---|---|---|---|---|---|
| | ACC | CAA | NMI | ARI | ACC | CAA | NMI | ARI |
| SeCu | 53.1±0.9* | 58.1±1.0* | 57.1±0.2* | 41.3±0.3* | 39.0±0.4* | 37.7±0.5* | 40.9±0.4* | 25.0±0.2* |
| LFSS | 54.5±1.8* | 59.1±1.6* | 57.9±0.1* | 42.0±0.3* | 41.9±0.7* | 40.6±0.4* | 46.0±0.3 | 28.2±0.1* |
| ConMix | 52.2±1.0* | 56.7±1.2* | 56.7±0.5* | 40.4±0.6* | 40.8±0.4* | 38.6±0.2* | 43.6±0.1* | 26.7±0.2* |
| MiniClustering | **65.1±0.4** | **61.2±0.1** | **63.6±0.3** | **57.2±0.4** | **43.8±0.5** | **41.5±0.4** | **46.3±0.8** | **30.1±0.4** |

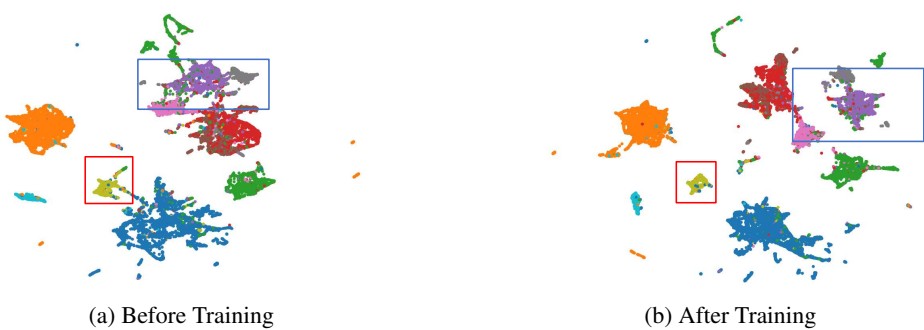

(a) Before Training                              (b) After Training

Figure 6: Visualization of learned embeddings before and after training on CIFAR-10 (imbalance ratio = 10). The class colors are ordered from highest to lowest sample count as follows: blue, orange, green, red, purple, brown, pink, gray, dark yellow, light blue.

# I   VISUALIZATIONS OF THE LEARNED EMBEDDINGS

We present the encoder output embeddings before and after training with MiniClustering. We use UMAP (McInnes et al., 2018) to reduce the high-dimensional features of CIFAR-10 (imbalance ratio = 10) to two dimensions and visualize them as scatter plots in Figure 6. It can be observed that after training, the embeddings of tail classes have become more distinguishable from those of head classes. For example, the dark yellow tail class is now farther away from the blue head class (red box). Additionally, the pink and gray tail-class samples, which were relatively close to the purple head class before training, have become more separated from it after training (blue box).

# J   MINICLUSTERING ON TEXT DATASETS

Our method can also be applied to non-image data, as its effectiveness does not rely on image-specific components. To demonstrate this, we conduct experiments on text datasets. Specifically, we construct imbalanced versions of the StackOverflow (Xu et al., 2017) and AgNews (Rakib et al., 2020) datasets with an imbalance ratio of 10, and train MiniClustering on them. Following (Zhang et al., 2021a), we use distilbert-base-nli-stsb-mean-tokens (Reimers and Gurevych, 2019) as the backbone for extracting features. We use WordNet (**?**) for synonym replacement as the data augmentation technique (Morris et al., 2020). As in the image experiments, we train both the target-cluster head and the mini-cluster head using the estimated class-specific weights. Since distilbert-base-nli-stsb-mean-tokens itself is a pre-trained model, we can directly use it for MiniClustering. To validate the effectiveness, we will use K-means on the embeddings output by distilbert-base-nli-stsb-mean-tokens before training as a baseline, and employ the self-labeling strategy with a threshold as a comparative method (i.e., only target-cluster head). We also reproduced ConMix (Li and Jia, 2025) on text datasets for comparison. The experimental results are shown in Table 10. Our method demonstrates a significant improvement compared to the baseline and the self-labeling approach, proving its effectiveness.The experimental results confirm the effectiveness of our approach in non-image scenarios.

Table 10: Performance comparison on text datasets (imbalance ratio = 10).

| Method | StackOverflow | | | | AgNews | | | |
|---|---|---|---|---|---|---|---|---|
| | ACC | CAA | NMI | ARI | ACC | CAA | NMI | ARI |
| Baseline | 57.7 | 49.9 | 51.5 | 43.4 | 53.1 | 50.1 | 31.2 | 25.0 |
| Self-labeling | 59.0 | 51.9 | 53.3 | 45.4 | 56.6 | 52.5 | 38.6 | 30.7 |
| ConMix | 58.0 | 49.1 | 53.4 | 45.5 | 55.6 | 54.4 | 38.4 | 31.1 |
| Ours | **65.3** | **57.7** | **59.0** | **55.8** | **58.7** | **55.3** | **39.7** | **32.7** |

Table 11: Clustering results (in percent %) of various methods on balanced datasets and imbalance datasets with an imbalance ratio = 10.

| Datasets | CIFAR-10 | | | | | | CIFAR-20 | | | | | | Average | | |
|---|---|---|---|---|---|---|---|---|---|---|---|---|---|---|---|
| Data Type | Balanced | | | Imbalanced | | | Balanced | | | Imbalanced | | | | | |
| Metric | ACC | NMI | ARI | ACC | NMI | ARI | ACC | NMI | ARI | ACC | NMI | ARI | ACC | NMI | ARI |
| IDFD | 81.5 | 71.1 | 66.3 | 47.5 | 48.4 | 33.1 | 42.5 | 42.6 | 26.4 | 28.7 | 28.6 | 15.1 | 50.1 | 47.7 | 35.3 |
| Propos | 91.6 | 85.1 | 83.5 | 46.1 | 52.5 | 34.2 | 57.8 | 58.2 | 42.3 | 36.8 | 40.1 | 22.5 | 58.1 | 59.0 | 45.6 |
| ConMix | 80.9 | 70.7 | 65.6 | 53.3 | 57.1 | 40.8 | 46.0 | 45.5 | 29.8 | 41.7 | 43.6 | 27.0 | 55.4 | 54.2 | 40.8 |
| MiniClustering | 91.1 | 84.5 | 82.5 | 64.6 | 63.9 | 56.7 | 54.6 | 57.6 | 40.5 | 44.3 | 47.2 | 30.4 | 63.7 | 63.3 | 52.5 |

## K  CLUSTERING PERFORMANCE ON BALANCED DATASETS

To evaluate overall performance of the proposed MiniClustering, we trained and evaluated our model on the balanced datasets CIFAR-10 and CIFAR-20. We used the combined set of the training and test splits of these datasets, while keeping all other experimental settings consistent with those described in the main text. The experimental results are shown in Table 11.

We compare our method with recent state-of-the-art balanced deep clustering approaches, including IDFD (Tao et al., 2020) and ProPos (Huang et al., 2023), as well as with ConMix (Li and Jia, 2025), a novel long-tailed deep clustering method.

Experimental results demonstrate that our method still achieves strong performance on balanced datasets, outperforming both IDFD and ConMix. Although it falls slightly short of ProPos, the performance gap is acceptable, as our primary focus is on addressing long-tailed deep clustering, where our method excels. Also, we compute the average of all metrics across the four datasets (long-tailed and balanced CIFAR-10 and CIFAR-20). It can be seen that our method outperforms other approaches, demonstrating superior comprehensive performance.

## L  PROOF AND ANALYSIS OF THEOREM 1

We assume that samples from $K$ classes follow a long-tailed distribution and are clustered into $M$ mini-clusters. The purity of in each mini-cluster is at least $\rho$. Let $S_{\min}$ and $S_{\max}$ be the minimum and maximum numbers of samples in mini-clusters, respectively. A mini-cluster is said to be dominated by class $k$ if its purity with respect to class $k$ is at least $\rho$. Let $m_k$ denote the number of mini-clusters dominated by class $k$. Let $\epsilon_k$ denote the number of samples from class $k$ that lie outside the mini-clusters dominated by class $k$. Let $N_k$ denote the number of samples in class $k$. Then, we can derive that

$$N_k = \sum_{i \in D_k} \rho_i S_i \geq \rho \sum_{i \in D_k} S_i \geq \rho m_k S_{\min},$$

where $D_k$ denotes the set of indices of mini-clusters dominated by class $k$, $\rho_i$ denotes the purity of mini-cluster $i$, which is not less than $\rho$, and $S_i$ denotes the size of mini-cluster $i$. Therefore, we can compute an upper bound for $m_k$ as follow:

$$m_k \leq \frac{N_k}{\rho S_{\min}},$$

and a larger $N_k$ implies a larger upper bound for $m_k$.

Also, we can assume that all mini-clusters in $D_k$ contain only samples from class $k$ and derive the upper bound of $N_k$:

$$N_k = \sum_{i \in D_k} \rho_i S_i + \epsilon_k \leq m_k S_{\max} + \epsilon_k,$$

and obtain the lower bound of $m_k$:

$$m_k \geq \frac{N_k - \epsilon_k}{S_{\max}}.$$

For two classes $i$ and $j$, $D_i > D_j$ means that the number of samples in class $i$ is greater than that in class $j$. When the lower bound of $m_i$ is greater than the upper bound of $m_j$, it necessarily follows that $m_i > m_j$. Thus, when $\frac{N_i - \epsilon_i}{S_{\max}} > \frac{N_j}{\rho S_{\min}}$, we can conclude a sufficient condition for $m_i > m_j$:

$$\rho > \frac{N_j S_{\max}}{(N_i - \epsilon_i) S_{\min}}.$$

To facilitate analysis, we assume $S_{\max} \approx S_{\min}$, meaning that the sizes of all mini-clusters are approximately equal. Then we obtain that $\rho > \frac{N_j}{N_i - \epsilon_i}$. Therefore, if $\rho$ is too small or $\epsilon_i$ is too large, it may lead to the failure of $m_i > m_j$ when $N_i > N_j$.

Moreover, both the minimum purity $\rho$ of mini-clusters and the number $\epsilon_i$ of class $i$ samples located outside the mini-clusters dominated by class $i$ depend on the quality of the embeddings. If the embedding quality is low and the model struggles to consistently group samples of the same class into the same mini-clusters, the above property may fail to hold. Therefore, in MiniClustering, we use a model pre-trained with unsupervised representation learning to ensure a reasonable embedding distribution.

