# OpenReview forum: "Mini-cluster Guided Long-tailed Deep Clustering"
_ICLR.cc/2026/Conference — ICLR 2026 Poster_

### Official Review · Reviewer_dtmG · 2025-10-26

**Soundness:** 2
**Presentation:** 3
**Contribution:** 3
**Rating:** 6
**Confidence:** 3

**Summary:**

The paper introduces a novel method to cluster longt-tailed datasets where the class distributions are imbalanced. The proposed method, called MiniClustering, enables class re-weighting in unsupervised learning(clustering). The network uses two heads
* A target-cluster head ($f_t$) that predicts the desired number of clusters ($K$).
* A mini-cluster head ($f_m$) that predicts a much larger number of clusters ($M$).
By analyzing how many mini-clusters are associated with each target-cluster, the method estimates which target-clusters represent "head" vs. "tail" classes. It then calculates a weight for each target-cluster: clusters associated with many mini-clusters (head classes) get a smaller weight, and those with few (tail classes) get a larger weight. This weight is used in a re-weighted self-training loss ($L_r$) for the target-cluster head. The full model is trained with this loss, a self-training loss for the mini-cluster head ($L_m$), and a similarity alignment loss ($L_s$) to keep the two heads synchronized.

**Strengths:**

1. The papers primary contribution is a functional mechanism to perform class balancing in a unsupervised setting. Which is mostly limited to supervised learning until now.
2. The method is rather flexible in terms of representation learning framwork. The authors demonstrate this with BYOL,SimCLR and MoCO
3. The authors provide a comprehensive ablation study to justify each component of the method and providing some insides how these components influence the clustering quality

**Weaknesses:**

1. Hyperparameter Sensitivity, Minicluster introduces 4 new hyperparametes that lack a unsupervised tuning strategy:
* Number of mini-clusters
* threshold for assigning a mini-cluster to a target-cluster
* $\alpha$ and $\beta$ as trade-off weights for the losses. This parameters are quite significant as shown in figure 5.
2. The MaxiClustering variant proposed in the Appendix feels a post patch that contradicts some of the core logic of the method.
* MiniClustering ($M > K$) is based on the premise that head classes spread across many mini-clusters.
* MaxiClustering ($M < K$) is based on the inverted premise that tail classes spread across many maxi-clusters.
MaxiClustering inverts the re-weighting logic, hypothesizing that tail classes are associated with more maxi-clusters. This seems to be a direct contradiction of the "phenomenon" that motivated the entire paper.
3. Implicit Assumption of K. This is a common limitation/weakness of most SoTA e.g. Secu and SCAN (ref. in the paper) but a weakness nonetheless

**Questions:**

1. How can the hyperparameters be tuned in a real-world, fully unsupervised scenario?
2. How robust is the core assumption (Section 3, Phenomenon 3) across different types of data and encoders?
3. Also related to 2., since there is a MiniClustering and MaxiClustering. What is the theoretical basis for this inversion, and does it imply the Phenomenon 3 is just a heuristic dependent on the $M/K$ ratio?
4. How much of the performance gain comes from the pre-training versus the re-weighting? How much would the baseline improve if it would continue the 200 epochs of clustering?

---

> ### Author Response · Authors · 2025-11-24
> **Response to Reviewer dtmG (1/3)**
>
> Thank you for your careful review and constructive comments. Below is our response:
>
> > ### Weakness 1 and Question 1: Hyperparameter sensitivity and how can the hyperparameters be tuned in a real-world, fully unsupervised scenario.
>
> **Response**: We agree with the reviewer that hyperparameter tuning is a common challenge faced by all unsupervised learning tasks. **We hope to leverage empirical insights gained from current datasets to derive parameter guidelines that can be applied in fully unsupervised settings**. So we conduct experiments on CIFAR-10 and STL-10 with an imbalance ratio of 10, using various hyperparameter settings. We adopt a one-variable-at-a-time approach: when one hyperparameter is varied for analysis, all others remain fixed and consistent with those used in the main results. The results are in **Appendix D** of the paper. Our method consistently outperforms the baseline across nearly all hyperparameter combinations, with only one exception. Clustering performance remains stable within a reasonable range under different hyperparameter settings. To achieve better performance, the following recommendations are worth considering:
>
> **The number of mini-clusters $M$**. We suggest selecting a value greater than the actual number of classes but no more than four times that number. When $M$ is set equal to the actual number of classes, the resulting partitioning closely resembles that of the target clusters, thereby failing to leverage the benefits offered by mini-clusters for performance improvement. For instance, on STL-10, setting $M$=10 leads to a noticeable performance drop. Conversely, when $M$ is too large, the overly fine-grained partitioning hinders the encoder from learning representations well-suited for predicting target clusters. For example, on CIFAR-10, performance degrades when $M$ is set to 50 or 60.
>
> **Mini-cluster assignment threshold δ**. We find it relatively stable across various settings and recommend using 0.3 or 0.5, which assigns more mini-clusters to each target cluster. And a more inclusive choice of δ is also beneficial when embedding quality is poor.
>
> **Trade-off parameters α and β**. α has a more pronounced impact on performance. For instance, CIFAR-10 tends to favor smaller values of α, while STL-10 prefers larger values. This behavior may be related to the intrinsic characteristics of the datasets. Therefore, we suggest using 0.5 as a default setting for α, if no prior knowledge about the dataset is available. In contrast, the choice of β exhibits relatively stable performance across different values, with most settings yielding comparable results.

---

> > ### Author Response · Authors · 2025-11-24
> > **Response to Reviewer dtmG (2/3)**
> >
> > > ### Weakness 2 and Question 3: Regarding the inversion of weight estimation in MaxiClustering.
> >
> > **Response**: The primary reason we reversed the weight estimation logic in MaxiClustering stems from several key considerations. First, the weight estimation strategy from MiniClustering in Eq. (2) uses the number of samples in the maxi-cluster as the denominator to calculate whether the assignment threshold has been exceeded. This fails in the maxi-cluster scenario because the maxi-cluster often contains more samples than the target-cluster, making it difficult to meet the assignment threshold. As a result, many target-clusters end up not being assigned any maxi-cluster in this scheme.
> >
> > Therefore, we set the denominator in the weighting formula to the number of samples in each target cluster. Specifically, a maxi-cluster is assigned to a target cluster if more than a certain proportion of **that target cluster's** samples belong to the maxi-cluster. Under this formulation, the target cluster is easier to be assigned a certain maxi-cluster.
> >
> > In fact, most target-clusters are assigned only one maxi-cluster. This is because, when the number of classes is large, even though a single head class contains far more samples than tail classes, it is still unlikely to dominate the entire dataset. Cases where a target-cluster is assigned to multiple maxi-clusters are typically caused by clustering inconsistency, indicating that the embeddings within that target-cluster are not sufficiently compact and thus get scattered across multiple maxi-clusters. Meanwhile, since the target-clusters of tail classes are generally smaller than those of head classes, they are easier to be assigned more maxi-clusters under the same threshold when such clustering inconsistency happens. Therefore, clusters corresponding to tail classes are more likely to be assigned multiple maxi-clusters and should be given higher weights.
> >
> > For example, we cluster Tiny ImageNet (200 classes) into 100 maxi-clusters and assign maxi-cluster memberships using a threshold of 0.3. We divide the 200 classes into four groups of 50 classes each, ranked from highest to lowest by number of samples, resulting in groups with varying degrees of long-tailedness. The average number of assigned maxi-clusters per group is [1.56, 1.58, 1.86, 2.32], and tail classes are probabilistically more likely to be assigned a larger number of maxi-clusters.
> >
> > Our explanation in **Appendix F** was previously too brief so we have revised this section to make it clearer and easier to understand.
> >
> > > ### Weakness 3: Implicit Assumption of the number of classes $K$.
> >
> > **Response**: Admittedly, existing methods often assume that the number of clusters is known. We adopt the same experimental setup as our compared methods for a fair comparison. We agree that this is a weakness for most of the current deep clustering methods.
> >
> > > ### Question 2: How robust is the core assumption (Section 3, Phenomenon 3) across different types of data and encoders?
> >
> > **Response**: Indeed, the core assumption (Phenomenon 3, Section 3) generalizes well across different data modalities and encoder designs. We validate this through experiments on a text dataset.Following [R1], we use distilbert-base-nli-stsb-mean-tokens as the encoder, which is a Transformer-based BERT variant specifically designed for extracting textual embeddings, which differs from the ResNet architecture used in our image-based experiments.
> > We evaluate on the StackOverflow text dataset, which originally contains 20 balanced classes. As described in the main text, we construct a long-tailed version with an imbalance ratio of 10, designating the first 9 classes as head classes (accounting for approximately 70% of all samples) and the remaining 11 as tail classes.
> > Mirroring the setup of Phenomenon 3, we examine the average number of mini-clusters assigned to head versus tail classes under varying values of the mini-cluster count $M$ and assignment threshold $\delta$. Results are presented below
> >
> > $M$=50, $\delta$ varies:
> >
> > |     | Head   | Tail   |
> > |-----|--------|--------|
> > | 0.3 | 2.6667 | 2.4545 |
> > | 0.5 | 2.4444 | 1.8182 |
> > | 0.7 | 2.0000 | 1.5455 |
> >
> > $\delta$=0.5, $M$ varies:
> >
> > |    | Head   | Tail    |
> > |----|--------|---------|
> > | 40 | 1.8889 | 1.8182  |
> > | 50 | 2.4444 | 1.8182  |
> > | 60 | 2.8889 | 1.7273  |
> >
> > Our results confirm that this phenomenon persists across different data types and encoder architectures.
> >
> > We have conducted MiniClustering experiments on text datasets in Appendix K, which shows that the proposed MiniClustering is also applicable to text datasets..
> >
> > [R1] Supporting Clustering with Contrastive Learning, NAACL, 2021.

---

> > > ### Author Response · Authors · 2025-11-24
> > > **Response to Reviewer dtmG (3/3)**
> > >
> > > > ### Question 3: How much would the baseline improve if it would continue the 200 epochs of training?
> > >
> > > **Response**: Thank you for your question. We further trained the BYOL baseline for 200 additional epochs, and its performance improved in most cases. But it still falls short of our method. We conducted experiments on CIFAR-10, CIFAR-20, and STL-10 with an imbalance ratio of 10. The results are as follows:
> > >
> > > CIFAR-10:
> > >
> > > |                   | ACC  | CAA  | NMI  | ARI  |
> > > |-------------------|------|------|------|------|
> > > | BYOL (800 epoch)  | 51.9 | 55.2 | 56.3 | 41.7 |
> > > | BYOL (1000 epoch) | 54.6 | 58.7 | 57.3 | 42.1 |
> > > | MiniClustering    | **64.6** | **61.4** | **63.9** | **56.7** |
> > >
> > > CIFAR-20:
> > >
> > > |                   | ACC  | CAA  | NMI  | ARI  |
> > > |-------------------|------|------|------|------|
> > > | BYOL (800 epoch)  | 39.8 | 37.1 | 41.9 | 25.0 |
> > > | BYOL (1000 epoch) | 40.7 | 37.9 | 42.0 | 26.7 |
> > > | MiniClustering    | **44.3** | **41.5** | **47.2** | **30.4** |
> > >
> > > STL-10:
> > >
> > > |                   | ACC  | CAA  | NMI  | ARI  |
> > > |-------------------|------|------|------|------|
> > > | BYOL (800 epoch)  | 41.2 | 41.8 | 46.8 | 29.6 |
> > > | BYOL (1000 epoch) | 44.9 | 37.4 | 48.8 | 31.4 |
> > > | MiniClustering    | **52.0** | **54.4** | **50.8** | **39.1** |
> > >
> > > These results can further demonstrate the effectiveness of our approach.

---

> ### Author Response · Authors · 2025-11-26
> **Looking forward to your further feedback**
>
> Dear Reviewer **dtmG**
>
> Thank you for taking the time to review our manuscript and for your valuable feedback and recognition. We have carefully addressed all the comments and concerns raised, as reflected in our detailed responses and the revised manuscript and supplementary material.
>
> We are looking forward to your further feedback.
>
> Best regards,
>
> The Authors

---

### Official Review · Reviewer_QMja · 2025-10-28

**Soundness:** 2
**Presentation:** 3
**Contribution:** 2
**Rating:** 4
**Confidence:** 4

**Summary:**

This paper addresses the long-tailed distributions in deep clustering. Since existing deep clustering methods assume class-balanced data, they struggle with real-world imbalanced scenarios. The authors propose MiniClustering that introduces an auxiliary mini-cluster head to generate finer-grained clusters than the final target number. These mini-clusters are used to estimate class-specific weights in an unsupervised manner, enabling re-weighted training analogous to supervised long-tailed learning. The framework includes three loss components: mini-cluster self-training loss, re-weighted target-cluster loss, and a similarity alignment loss to prevent desynchronization between clustering heads. Extensive experiments on CIFAR-10, CIFAR-20, STL-10, Tiny ImageNet, and ImageNet-LT, across varying imbalance ratios, show that MiniClustering consistently outperforms state-of-the-art baselines on multiple metrics.

**Strengths:**

1. The dual-head design is intuitive. It allows the model to approximate class-wise importance without needing labels, which is both elegant and practical in clustering scenarios.
2. The experimental results are strong and well-supported. Visualizations and benchmarks clearly show the method works better than existing ones across a variety of datasets and imbalance levels.

**Weaknesses:**

1. The main novelty of the paper lies in identifying the relationship between mini-clusters and target clusters, but these observations do not appear entirely original. The notion of mini-clusters has existed for years, with similar ideas explored in works like IIC and PICA. More specifically:

   a. Phenomena 1 and 3 would benefit from more conclusive, clearly stated takeaways rather than descriptive observations.

   b. Phenomenon 2 (“Mini-clusters can enhance purity”) is not very convincing, since purity naturally increases when clusters are subdivided. In the extreme case of infinite subdivisions, purity can trivially reach 1, which does not truly reflect clustering quality.

   c. Phenomenon 3 does not directly correspond to estimating class probabilities. The analysis is limited to comparing the relative proportions of head versus tail classes, without demonstrating whether this distinction can reliably separate them.

2. The results on balanced datasets are missing. It would be informative to see comparisons under standard settings—for example, relative to the results in Table 2 of ProPos. While this does not affect the core focus on long-tailed distributions, it would help situate the method’s overall competitiveness.
3. The similarity alignment loss resembles a second-order alignment loss introduced in *Graph Matching with Bi-level Noisy Correspondence*. A proper citation and discussion of this connection would strengthen the methodological grounding.

**Questions:**

Please refer to weaknesses.

---

> ### Author Response · Authors · 2025-11-24
> **Response to Reviewer QMja (1/2)**
>
> Thank you for raising these questions. We have carefully considered them and provide the following response.
>
> > ### Weakness 1: Similar ideas explored in works like IIC and PICA.
>
> **Response**: Indeed, both IIC and PICA employ an over-clustering head to obtain finer-grained predictions. However, in their frameworks, the over-clustering head is primarily used as an auxiliary objective to better train the feature encoder, and **it does not establish a direct connection with the target clustering head**. In contrast, our method leverages mini-cluster assignments to estimate the appropriate weight for each target cluster, thereby effectively addressing the challenging long-tailed clustering problem.
>
> > ### Weakness 1 a: Phenomena 1 and 3 would benefit from more conclusive, clearly stated takeaways rather than descriptive observations.
>
> **Response**: Thank you for your suggestion. We have revised the manuscript accordingly, using more conclusive takeaways instead of descriptive observations.
>
> Phenomena 1: Head classes occupy more embedding space than tail classes.
>
> Phenomena 3: Head classes are assigned more mini-clusters than tail classes.
>
> > ### Weakness 1 b: Phenomenon 2 (“Mini-clusters can enhance purity”) is not very convincing.
>
> **Response**: Indeed, using a larger number of clusters will inevitably increase intra-cluster purity. However, please note that our Phenomenon 2 is intended to motivate the use of mini-clusters. By obtaining finer-grained and purer clusters, we can better support Phenomenon 3 and explain how mini-cluster assignments can be used to estimate the degree of long-tailedness. We do not claim that higher intra-cluster purity alone necessarily indicates better clustering quality. In fact, as discussed in the hyperparameter analysis (Appendix D), an excessively large number of mini-clusters can actually degrade performance, because the resulting partitioning diverges significantly from the target cluster structure.
>
> > ### Weakness 1 c: Phenomenon 3 does not directly correspond to estimating class probabilities.
>
> **Response**: We can provide a more detailed explanation here. To analyze the relationship between the number of mini-clusters assigned to each target class and its corresponding sample count, we conduct experiments on CIFAR-10 with an imbalance ratio of 10. Consistent with the main text, we use pre-trained BYOL, SimCLR, and MoCo models, and compute how many mini-clusters each true class occupies. We set the number of mini-clusters $M$ to 40 and the threshold $\delta$ to 0.3, consistent with the experimental setup described in the paper. The results are shown in **Appendix H**. The ground-truth class indices from 0 to 9 represent classes ordered from head to tail (i.e., in decreasing order of sample counts). **We observe that the number of mini-clusters occupied by each true class correlates with its degree of long-tailedness, where head classes are assigned more mini-clusters than tail classes**. Thus, mini-cluster assignments reflect the probabilities of target classes.

---

> ### Author Response · Authors · 2025-11-24
> **Response to Reviewer QMja (2/2)**
>
> > ### Weakness 2: Experimental results on balanced datasets.
>
> **Response**: Thank you for your suggestion. We trained and evaluated our model on the balanced datasets CIFAR-10 and CIFAR-20. We used the combined set of the training and test splits of these datasets, while keeping all other experimental settings consistent with those described in the main text. The experimental results are shown below:
>
> Balanced CIFAR-10:
>
> |        | ACC  | NMI  | ARI  |
> |--------|------|------|------|
> | IDFD   | 81.5 | 71.1 | 66.3 |
> | Propos | 91.6 | 85.1 | 83.5 |
> | ConMix | 80.9 | 70.7 | 65.6 |
> | Ours   | 91.1 | 84.5 | 82.5 |
>
> Imbalanced CIFAR-10:
>
> |        | ACC  | NMI  | ARI  |
> |--------|------|------|------|
> | IDFD   | 47.5 | 48.4 | 33.2 |
> | Propos | 46.1 | 52.5 | 34.2 |
> | ConMix | 53.3 | 57.1 | 40.7 |
> | Ours   | 64.6 | 63.9 | 56.7 |
>
>
>
> Balanced CIFAR-20:
>
> |        | ACC  | NMI  | ARI  |
> |--------|------|------|------|
> | IDFD   | 42.5 | 42.6 | 26.4 |
> | Propos | 57.8 | 58.2 | 42.3 |
> | ConMix | 46.0 | 45.5 | 29.8 |
> | Ours   | 54.6 | 57.6 | 40.5 |
>
> Imbalanced CIFAR-20:
>
> |        | ACC  | NMI  | ARI  |
> |--------|------|------|------|
> | IDFD   | 28.7 | 28.6 | 15.1 |
> | Propos | 36.8 | 40.1 | 22.5 |
> | ConMix | 41.3 | 43.6 | 27.0 |
> | Ours   | 44.3 | 47.2 | 30.4 |
>
> Average Performance of four datasets:
>
> | Method | ACC  | NMI  | ARI  |
> |--------|------|------|------|
> | IDFD   | 50.1 | 47.7 | 35.3 |
> | Propos | 58.1 | 59.0 | 45.6 |
> | ConMix | 55.4 | 54.2 | 40.8 |
> | Ours   | 63.7 | 63.3 | 52.5 |
>
> We compare our method with recent state-of-the-art balanced deep clustering approaches, including IDFD and ProPos, as well as with ConMix, a novel long-tailed deep clustering method.
>
> Experimental results demonstrate that our method still achieves strong performance on balanced datasets, outperforming both IDFD and ConMix. Although it falls slightly short of ProPos, the performance gap is acceptable, as our primary focus is on addressing long-tailed deep clustering, where our method excels. Also, We compute the average of all metrics across the four datasets (long-tailed and balanced CIFAR-10 and CIFAR-20 combined). It can be seen that our method outperforms other approaches on average, demonstrating superior comprehensive performance. We have added content regarding performance on balanced datasets in **Appendix L**.
>
> > ### Weakness 3: Citation and discussion of the related second-order alignment approach.
>
>
> **Response**: Thank you for your advice. We have carefully read _Graph Matching with Bi-level Noisy Correspondence_. Eq. (7) and (8) in the paper serve as loss functions to measure within-graph consistency and cross-graph consistency, respectively. Their mathematical formulations align with our similarity alignment loss, and both are designed to preserve the consistency of relational weights. We have explicitly cited and discussed this paper in our manuscript. Thank you for your suggestion.

---

> ### Author Response · Authors · 2025-11-26
> **Looking forward to your further feedback**
>
> Dear Reviewer **QMja**
>
> Thank you for taking the time to review our manuscript and for your valuable feedback. We have carefully addressed all the comments and concerns raised, as reflected in our detailed responses and the revised manuscript and supplementary material.
>
> We are looking forward to your further feedback.
>
> Best regards,
>
> The Authors

---

### Official Review · Reviewer_amJw · 2025-10-29

**Soundness:** 3
**Presentation:** 2
**Contribution:** 2
**Rating:** 4
**Confidence:** 4

**Summary:**

To address the problems of current deep clustering, where multiple assumptions about data distribution are unbalanced and existing long-tail learning methods rely on labels and do not apply to unsupervised scenarios, this paper proposes MiniClustering (a long-tail deep clustering method guided by mini-clustering). By introducing a dual clustering head (a target clustering head + a mini-clustering head, where the former outputs a prediction of the number of target clusters and the latter outputs a more fine-grained mini-cluster prediction), unsupervised category weight estimation is achieved based on the core phenomenon that "head class samples are associated with more mini-clusters and tail classes with fewer mini-clusters". Finally, its effectiveness is verified across multiple datasets, including CIFAR-10/20 and STL-10.

**Strengths:**

1. The paper proposes an architecture of "target clustering head + mini clustering head" that estimates weights through the collaboration of clustering heads with different prediction granularities. This differs from the traditional single clustering head design and provides a new technical path for unsupervised reweighting.
2. The algorithm still demonstrated reliable performance even in scenarios with extreme imbalances and large-scale data.

**Weaknesses:**

1. The proposed MiniClustering relies on several key hyperparameters that need to be manually tuned across different datasets and imbalance ratios. It lacks automated adaptation capabilities, which increases the threshold for practical application.
2. All experiments were based on image data and did not involve non-image modalities such as text, audio, and video. Due to significant differences in data distribution and embedding characteristics across modalities, it is impossible to determine whether the "mini-clustering mechanism" of MiniClustering applies to non-image scenarios.

**Questions:**

1. Are there any rules for designing the number of mini-clustering heads, and what impact does it have on the stability of the model?
2. How does batch size affect model performance? In particular, how does the model perform when the batch size is smaller than the total number of classes?

---

> ### Author Response · Authors · 2025-11-24
> **Response to Reviewer amJw (1/2)**
>
> Thank you for your valuable feedback. We have carefully considered your comments and provided our answers below.
>
> > ### Weakness 1 and Question 1: About several key hyperparameters and rules for designing the number of mini-clsuters.
>
> **Response**: Our method indeed involves multiple hyperparameters. In Appendix D of our initial submission, we provided a hyperparameter analysis. Here, we supplement additional experiments and analyses to offer a more detailed explanation. To evaluate the impact of these hyperparameters on performance, we conduct experiments on CIFAR-10 and STL-10 with an imbalance ratio of 10, using various hyperparameter settings. We adopt a one-variable-at-a-time approach: when one hyperparameter is varied for analysis, all others remain fixed and consistent with those used in the main results. We present the results in **Appendix D** of the paper. Our results show that although performance varies across different hyperparameter settings, it remains within a reasonable range. **Most importantly, our method consistently outperforms the baseline** across the vast majority of hyperparameter combinations, except in one particular hyperparameter configuration. This demonstrates that our method maintains its effectiveness across a wide range of hyperparameter configurations.
>
> Regarding the specific design rule of the number of mini-clusters $M$, we suggest selecting a value greater than the actual number of classes but no more than four times that number. When $M$ is set equal to the actual number of classes, the resulting partitioning closely resembles that of the target clusters, thereby failing to leverage the benefits offered by mini-clusters for performance improvement. For instance, on STL-10, setting $M$=10 leads to a noticeable performance drop. Conversely, when $M$ is too large, the overly fine-grained partitioning hinders the encoder from learning representations well-suited for predicting target clusters. For example, on CIFAR-10, performance degrades when $M$ is set to 50 or 60.
>
> We have updated the **Appendix D** in the manuscript to better elaborate on the hyperparameter analysis.
>
> > ### Weakness 2: Applying to non-image scenarios.
>
> **Response**: Our method can also be applied to non-image data, as its effectiveness does not rely on image-specific components. To demonstrate this, we conduct experiments on two text datasets. Specifically, we construct imbalanced versions of the StackOverflow and AgNews datasets with an imbalance ratio of 10, and train MiniClustering on them. Following [R1], we use distilbert-base-nli-stsb-mean-tokens as the backbone for extracting features. We use WordNet for synonym replacement as the data augmentation technique. As in the image experiments, we train both the target-cluster head and the mini-cluster head using the estimated class-specific weights. Since distilbert-base-nli-stsb-mean-tokens itself is a pre-trained model, we can directly use it for MiniClustering. To validate the effectiveness, we will use K-means on the embeddings output by distilbert-base-nli-stsb-mean-tokens before training as a baseline, and employ the self-labeling strategy with a threshold as a comparative method (i.e., only target-cluster head). We also reproduced ConMix on text datasets for comparison. The experimental results are shown below:
>
> | StackOverflow | ACC  | CAA  | NMI  | ARI  |
> |---------------|------|------|------|------|
> | Baseline      | 57.7 | 49.9 | 51.5 | 43.4 |
> | Self-labeling | 59.0 | 51.9 | 53.3 | 45.4 |
> | ConMix        | 58.0 | 49.1 | 53.4 | 45.5 |
> | Ours          | **65.3** | **57.7** | **59.0** | **55.8** |
>
>
> | AgNews        | ACC  | CAA  | NMI  | ARI  |
> |---------------|------|------|------|------|
> | Baseline      | 53.1 | 50.1 | 31.2 | 25.0 |
> | Self-labeling | 56.6 | 52.5 | 38.6 | 30.7 |
> | ConMix        | 55.6 | 54.4 | 38.4 | 31.1 |
> | Ours          | **58.7** | **55.3** | **39.7** | **32.7** |
>
> Our method demonstrates a significant improvement compared to the baseline and the self-labeling approach, proving its effectiveness. The experimental results confirm the effectiveness of our approach in non-image scenarios. We have updated our manuscript and detailed experimental setting can be found in **Appendix K**.
>
> [R1] Supporting Clustering with Contrastive Learning, NAACL, 2021.

---

> > ### Author Response · Authors · 2025-11-24
> > **Response to Reviewer amJw (2/2)**
> >
> > > ### Question 2: The the impact of batch size on experimental performance.
> >
> > **Response**: To investigate the impact of batch size on performance, particularly when the batch size is smaller than the number of classes, we conduct experiments on imbalanced Tiny ImageNet with an imbalance ratio of 10. This dataset contains 200 classes, making it well-suited for our study. We fix the hyperparameters as M=300, δ=0.5, and α=β=1, and train models using batch sizes of 64, 128, 256, and 512. Notably, **both 64 and 128 are smaller than the total number of classes (200)**. For comparison, we also train models using the self-labeling strategy with these same four batch sizes as a baseline. The self-labeling method trains with cross-entropy loss using pseudo-labels selected by a threshold, thereby excluding the potential influence of batch size under mini-cluster assignment scenarios. The results are reported below:
> >
> > | Batch Size | Method         | ACC  | CAA  | NMI  | ARI  |
> > |------------|----------------|------|------|------|------|
> > | 64         | Self-labeling  | 19.2 | 12.8 | 36.2 | 8.8  |
> > | 64         | MiniClustering | 21.0 | 13.8 | 37.2 | 10.0 |
> > | 128        | Self-labeling  | 22.0 | 15.2 | 37.9 | 11.4 |
> > | 128        | MiniClustering | 22.5 | 15.5 | 37.8 | 10.4 |
> > | 256        | Self-labeling  | 23.5 | 16.6 | 38.8 | 12.2 |
> > | 256        | MiniClustering | 25.0 | 18.1 | 39.6 | 14.2 |
> > | 512        | Self-labeling  | 24.8 | 18.3 | 38.9 | 14.0 |
> > | 512        | MiniClustering | 25.3 | 18.8 | 40.3 | 14.1 |
> >
> > As observed, like most methods, our approach is somewhat affected by batch size. **However, it consistently outperforms the self-labeling baseline across all settings, demonstrating the effectiveness and robustness of our method**. Even when the batch size is smaller than the number of classes, performance degradation is modest and does not cause our method to fall below the self-labeling baseline. This demonstrates that, although our method employs class-specific processing, it does not strictly require the batch size to be larger than the number of classes.

---

> ### Author Response · Authors · 2025-11-26
> **Looking forward to your further feedback**
>
> Dear Reviewer **amJw**
>
> Thank you for taking the time to review our manuscript and for your valuable feedback. We have carefully addressed all the comments and concerns raised, as reflected in our detailed responses and the revised manuscript and supplementary material.
>
> We are looking forward to your further feedback.
>
> Best regards,
>
> The Authors

---

### Official Review · Reviewer_KT3s · 2025-10-29

**Soundness:** 2
**Presentation:** 3
**Contribution:** 3
**Rating:** 4
**Confidence:** 3

**Summary:**

The paper introduces a novel method for unsupervised long-tailed deep clustering to address the challenges posed by imbalanced data distributions. The proposed method attempts to mitigate the bias towards majority classes by introducing an auxiliary "mini-clustering" head which over-clusters the data (into many more clusters than the target number) to estimate class-specific weights for a re-weighted self-training loss.

**Strengths:**

- The idea of estimating imbalance through over-clustering (mini-clusters) is creative and conceptually plausible.
- The loss formulation is clearly described and easy to follow.
- Comprehensive comparisons across multiple datasets and imbalance ratios, with also ablation studies to validate design choices. The addition of the maxi-clustering variant is also appreciated.
- The paper is generally well-written and easy to read.

**Weaknesses:**

- While the method is motivated by empirical observations (Figures 1–3), there is no theoretical justification connecting the number of mini-clusters a target cluster occupies to its degree of long-tailedness. The proposed weight computation (Eq. 2) is purely heuristic, and the paper offers no probabilistic or geometric reasoning explaining why these weights should correlate with true class frequencies.
- The method’s weight estimation depends on pseudo-labels from two heads — both trained simultaneously. Early in training, these pseudo-labels are noisy, which could cause unstable or biased weight updates. No mitigation (e.g., confidence annealing, temporal ensembling, or delayed weighting) is described, and the paper does not analyze the effect of pseudo-label noise on convergence.
- The method relies on a good quality of the mini-clustering head to estimate class frequencies, but there is no analysis of how sensitive the method is to errors in this estimation.
- The experimental results, while comprehensive, lack statistical significance analysis. It is unclear whether the observed improvements over baselines are statistically significant or could be due to random variation.
- Figure 2 and 3 are a bit too small to read comfortably.

**Questions:**

- Can you provide any formal argument or empirical correlation analysis linking the number of mini-clusters assigned to a target-cluster and the true class frequency?
- If some minority classes are not well separated in the mini-clustering, how does this affect the final performance?
- Could you provide standard deviations or confidence intervals for the results in Tables 1 and 2 to assess statistical significance?
- Could you provide some visualizations of the learned embeddings (e.g., t-SNE or UMAP plots) to illustrate how well the method separates classes, especially minority ones?

---

> ### Author Response · Authors · 2025-11-24
> **Response to Reviewer KT3s (1/3)**
>
> Thank you for your constructive comments. We have carefully considered your feedback and provided the response as below.
>
> > ### Weakness 1 and Question 1: Empirical Analysis of True Class Frequency and Mini-cluster Assignments.
>
> **Response**: Yes, we can surely provide **an empirical correlation analysis linking mini-cluster assignments and true class frequency**. We conduct experiments on CIFAR-10 with an imbalance ratio of 10. Consistent with the main text, we use pre-trained BYOL, SimCLR, and MoCo models, and compute how many mini-clusters each true class occupies. We set the number of mini-clusters $M$ to 40 and the threshold $\delta$ to 0.3, consistent with the experimental setup described in the paper. The results are shown in Appendix H of the paper. The ground-truth class indices from 0 to 9 represent classes ordered from head to tail (i.e., in decreasing order of sample counts). We observe that the number of mini-clusters occupied by each true class correlates with its degree of long-tailedness: head classes are assigned more mini-clusters than tail classes. This aligns with our intuition and the observations presented in the motivation section 3.1. We have revised the manuscript and added this content to **Appendix H**.
>
> > ### Weakness 2: Measures for handling highly noisy pseudo-labels in the early stages of training.
>
> **Response**: You suggest that pseudo-labels in the early stages of training contain substantial noise, and that our method does not employ any mitigation strategies. This is indeed a thoughtful and professional critique. However, it may stem from a slight misunderstanding of our training framework. **In fact, we have implemented multiple measures to ensure the reliability and effectiveness of the pseudo-labels**.
>
> In our approach, before performing mini-clustering and training the two clustering heads, we first pre-train the encoder using self-supervised representation learning frameworks such as BYOL. Therefore, we do not use pseudo-labels from the very beginning of training. Instead, pseudo-label-based training is built upon a reasonably well-learned representation model.
>
> Moreover, we initialize the weights of the clustering heads using cluster centroids obtained via K-means clustering on the pre-trained features, following [R1]. This effectively transfers knowledge from the pre-trained encoder to the clustering heads, providing them with a strong initialization prior for training.
>
> Additionally, during pseudo-label training, we apply a confidence threshold to filter samples, using only high-confidence predictions for updating the model. This strategy helps mitigate the impact of potential noise in the pseudo-labels.
>
> To further validate the effectiveness of above strategies, we computed the clustering performance of the pseudo-labels after threshold filtering at the beginning of training (epoch 0). These are the pseudo-labels that actually participate in backpropagation.
>
> The experimental results are shown below. It can be observed that, compared to the baseline (K-means on pretrained BYOL without threshold filtering), the **pseudo-label accuracy from the target head after threshold filtering is significantly improved, even before any training**. Considering the long-tailed distribution, under which all methods exhibit relatively low performance, this level of performance is already remarkably good.
>
> |                                             | ACC  | CAA  | NMI  | ARI  |
> |---------------------------------------------|------|------|------|------|
> | Baseline (all data)                         | 60.4 | 66.4 | 62.0 | 46.5 |
> | epoch 0 (selected data for pseudo labeling) | 71.9 | 67.5 | 83.3 | 65.7 |
>
> [R1] Semantic-enhanced image clustering, AAAI, 2023.

---

> > ### Author Response · Authors · 2025-11-24
> > **Response to Reviewer KT3s (2/3)**
> >
> > > ### Weakness 3 and Question 2: If some minority classes are not well separated in the mini-clustering, how does this affect the final performance?
> >
> > **Response**: In fact, the reason we adopt the mini-clustering strategy is **to separate tail classes (minority classes) from head classes with reasonably high purity in an unsupervised manner**. This is the fundamental principle of our work.
> >
> > If the mini-cluster approach is not adopted, and clustering is performed using the number of target classes as is commonly done nowadays, tail classes (minority classes) are forced to share a cluster with head classes, as revealed in Phenomenon 1 of Section 3. Here, we can provide a more detailed experiment to illustrate this point. We conduct experiments using BYOL pre-trained on CIFAR-10 (imbalance ratio = 10). We applied K-means to cluster the embeddings into 10 clusters. We further computed, for each of these clusters, (1) the purity of the target class within the cluster, and (2) the proportion of samples in the target class that belong to this certain cluster. We examine the four classes with indices 6 to 9, as they have the fewest samples and can be considered tail classes. The results are presented below:
> > | class | purity | proportion |
> > |-------|--------|-----------|
> > | 6     | 0.78   | 0.45      |
> > | 7     | 0.66   | 0.83      |
> > | 8     | 0.21   | 0.89      |
> > | 9     | 0.15   | 0.73      |
> >
> > It can be observed that when clustering with the target number of clusters, the purity of clusters corresponding to tail classes is relatively low, **indicating that these classes are not well separated from other categories. However, this issue can be effectively addressed by adopting the mini-cluster strategy**.
> >
> > To validate this claim, we also conduct experiments using BYOL pre-trained on CIFAR-10 (imbalance ratio = 10). But we applied K-means to cluster the embeddings into 40 mini-clusters. We used 0.3 as the threshold to count the number of mini-clusters occupied by each class. We compute, as above, the purity of each cluster and the proportion of samples in that cluster relative to the total number of samples in the corresponding class. We examine the cluster assignment results for the four tail classes with the smallest number of samples. The results are as follows:
> >
> > | class | purity | proportion |
> > |-------|--------|------------|
> > | 6     | 0.78   | 0.45       |
> > | 6     | 0.78   | 0.42       |
> > | 7     | 0.87   | 0.63       |
> > | 8     | 0.90   | 0.85       |
> > | 9     | 0.90   | 0.87       |
> >
> > We found that the four classes with the fewest samples (class indices 6 to 9) collectively occupied 5 mini-clusters. Specifically, class 6 occupied two mini-clusters, while each of the other three classes (7, 8, and 9) occupied one mini-cluster each. Also, we observed that the purity within each occupied mini-cluster was very high, and the majority of samples from each tail class were assigned to their respective mini-clusters. **This demonstrates that our approach can effectively seperate tail classes**.

---

> > > ### Author Response · Authors · 2025-11-24
> > > **Response to Reviewer KT3s (3/3)**
> > >
> > > > ### Weakness 4 and Question 3: Statistical significance analysis (weakness 4 and question 3)
> > >
> > > **Response**: To clarify, in the experimental results of Section 4.2, for all reproduced methods, we used the random seeds employed in their original code. For our method, we used the same random seed across different datasets to ensure a fair comparison. To demonstrate that the performance improvement is not due to random chance, we ran our method multiple times using different random seeds and computed the mean and standard deviations of its performance. For a fair comparison, we reproduced the most competitive methods, including SeCu, LFSS, and ConMix, for evaluation. Experiments were conducted on imbalanced versions of CIFAR-10 and CIFAR-20 with an imbalance ratio of 10. We repeated each experiment five times using the same random seeds for all methods. The results are presented below:
> > >
> > > CIFAR-10:
> > >
> > > |        | ACC       | CAA       | NMI       | ARI       |
> > > |--------|-----------|-----------|-----------|-----------|
> > > | SeCu   | 53.1±0.9* | 58.1±1.0* | 57.1±0.2* | 41.3±0.3* |
> > > | LFSS   | 54.5±1.8* | 59.1±1.6* | 57.9±0.1* | 42.0±0.3* |
> > > | ConMix | 52.2±1.0* | 56.7±1.2* | 56.7±0.5* | 40.4±0.6* |
> > > | Ours   | **65.1±0.4**  | **61.2±0.1**  | **63.6±0.3**  | **57.2±0.4**  |
> > >
> > > CIFAR-20:
> > >
> > > |        | ACC       | CAA       | NMI       | ARI       |
> > > |--------|-----------|-----------|-----------|-----------|
> > > | SeCu   | 39.0±0.4* | 37.7±0.5* | 40.9±0.4* | 25.0±0.2* |
> > > | LFSS   | 41.9±0.7* | 40.6±0.4* | 46.0±0.3  | 28.2±0.1* |
> > > | ConMix | 40.8±0.4* | 38.6±0.2* | 43.6±0.1* | 26.7±0.2* |
> > > | Ours   | **43.8±0.5**  | **41.5±0.4**  | **46.3±0.8**  | **30.1±0.4**  |
> > >
> > > As shown, our method consistently achieves significantly better performance in terms of mean values compared to other approaches, especially exhibiting a clear advantage on the CIFAR-10 dataset. We use the t-test for significance testing and mark with an asterisk (*) the metrics where our method is significantly better than other methods at the 95% confidence level. Our method **is significantly better** than the most competitive baseline in the majority of cases. We have updated the manuscript and included the above results in **Appendix I**.
> > >
> > >  > ### Question 4: Visualizations of the learned embeddings.
> > >
> > > **Response**: Thank you for your suggestion. We will show the encoder output embeddings before and after training with MiniClustering. We use UMAP to reduce the high-dimensional features of CIFAR-10 (imbalance ratio = 10) to two dimensions and visualize them in a scatter plot. The results are available in **Appendix J** of the paper. It can be observed that after training, the embeddings of tail classes have become more distinguishable from those of head classes. For example, the dark yellow tail class is now farther away from the light blue head class (red box). Additionally, the pink and gray tail-class samples, which were relatively close to the purple head class before training, have become more separated from it after training (blue box). We have updated the manuscript and added above content.
> > >
> > > > ### Weakness 5: Figure 2 and 3 are a bit too small to read comfortably.
> > >
> > > **Response**: Thank you for your reminder. In the revised paper, we have adjusted the manuscript layout to make Figures 2 and 3 easier to read.

---

> ### Author Response · Authors · 2025-11-26
> **Looking forward to your further feedback**
>
> Dear Reviewer **KT3s**
>
> Thank you for taking the time to review our manuscript and for your valuable feedback. We have carefully addressed all the comments and concerns raised, as reflected in our detailed responses and the revised manuscript and supplementary material.
>
> We are looking forward to your further feedback.
>
> Best regards,
>
> The Authors

---

> > ### Comment · Reviewer_KT3s · 2025-11-26
> > **post-rebuttal answer**
> >
> > Thank you for the detailed response and for the additional clarifications. The experiments on multiple seeds shows that the method is consistent. However, I don't think that some of my concerns were sufficiently addressed.
> >
> > The rebuttal largely reiterates the intuition already presented in the paper, without providing a deeper or more rigorous theoretical explanation. The core assumption, i.e. that the number of mini-clusters reliably reflects true class frequency, remains mostly empirica, lacking theoretical rigor that would be beneficial. There is no analysis of the conditions under which this relationship holds, how it depends on representation quality or cluster geometry, or when it may fail.
> >
> > Moreover, the authors repeatedly state that materials (Appendix H/J, enlarged figures, added experiments) have been added to the manuscript, but these changes are not visible to reviewers. As a result, it is impossible to verify whether the new content adequately addresses the concerns.
> >
> > In summary, as I already stated in the review, the idea of over-clustering is creative and conceptually plausible, but it still lacks a thorough theoretical analysis. Hence, my overall evaluation and score remain unchanged.

---

> > > ### Author Response · Authors · 2025-11-28
> > > **Response to post-rebuttal answer from Review KT3s (1/2)**
> > >
> > > Thank you for your further reply. Below is our response.
> > >
> > > Regarding the content that has been added to or modified in the paper, all the modifications and additions we mentioned have been incorporated into the manuscript (PDF version). You can find and download the updated version right where you viewed our original paper on OpenReview website. In particular, we have highlighted the revised or newly added content in blue to help you easily identify the changes. Below, we summarize the updates made in response to your comments:
> > >
> > > (1)	We provided an empirical correlation analysis linking mini-cluster assignments and true class frequency in **Appendix H**.
> > >
> > > (2)	We added statistical significance analysis in **Appendix I**.
> > >
> > > (3)	We added visualizations of the learned embeddings in **Appendix J**.
> > >
> > > (4)	We adjusted the manuscript layout to make Figures 2 and 3 easier to read in Section 3.1.
> > >
> > > For the newly added or modified content, you may refer either to our discussion in the paper itself or to our previous response to you. We have provided detailed analysis and explanations.
> > >
> > > We have devoted considerable effort to theoretically analyzing the relationship between the mini-cluster assignment and the ground-truth class frequencies. **As a result, we are now able to present a well-founded theoretical explanation demonstrating that, under appropriate conditions, classes with larger sample sizes are assigned more mini-clusters**.
> > >
> > > Specifically, we formalize this insight in a theorem in **Section 3.1** (Theoretical Analysis) of the revised paper and provide a detailed derivation, proof, and discussion in **Appendix M** of the revised paper. We clarify that the validity of Phenomenon 3 hinges on both the purity of mini-clusters and the quality of clustering, particularly, it relies on a well-structured embedding distribution. Conversely, poor-quality embeddings are more likely to violate the assumptions underlying this phenomenon, thereby causing Phenomenon 3 to fail.

---

> ### Author Response · Authors · 2025-11-28
> **Response to post-rebuttal answer from Review KT3s (2/2)**
>
> For your convenience, we also provide the theoretical proof and analysis here.
>
> We assume that samples from $K$ classes follow a long-tailed distribution and are clustered into $M$ mini-clusters. The purity of each mini-cluster is at least $\rho$. Let $S_{\text{min}}$ and $S_{\text{max}}$ be the minimum and maximum numbers of samples in mini-clusters, respectively. A mini-cluster is said to be dominated by class $k$ if its purity with respect to class $k$ is at least $\rho$. Let $m_k$ denote the number of mini-clusters dominated by class $k$. Let $\epsilon_k$ denote the number of samples from class $k$ that lie outside the mini-clusters dominated by class $k$. Let $N_k$ denote the number of samples in class $k$.
>
> Theorem 1. For classes $i$ and $j$, if the number of samples $N_i > N_j$, then the number of occupied mini-clusters $m_i > m_j$ holds when
> $$
> \rho > \frac{N_j S_{\max}}{(N_i - \varepsilon_i) S_{\min}}.
> $$
>
>
> In the following, we provide a detailed proof and analysis.
>
> First, we can derive that
> $$
> N_k = \sum_{i \in D_k} \rho_i S_i \geq \rho \sum_{i \in D_k} S_i \geq \rho m_k S_{\min},
> $$
> where $D_k$ denotes the set of indices of mini-clusters dominated by class $k$, $\rho_i$ denotes the purity of mini-cluster $i$ (which is not less than $\rho$), and $S_i$ denotes the size of mini-cluster $i$.
>
> Therefore, we can compute an upper bound for $m_k$ as follows:
> $$
> m_k \leq \frac{N_k}{\rho S_{\min}},
> $$
> and a larger $N_k$ implies a larger upper bound for $m_k$.
>
> Also, we can assume that all mini-clusters in $D_k$ contain only samples from class $k$ and derive the upper bound of $N_k$:
> $$
> N_k = \sum_{i \in D_k} \rho_i S_i + \epsilon_k \leq m_k S_{\max} + \epsilon_k,
> $$
> and obtain the lower bound of $m_k$:
> $$
> m_k \geq \frac{N_k - \epsilon_k}{S_{\max}}.
> $$
>
> For two classes $i$ and $j$, $N_i > N_j$ means that the number of samples in class $i$ is greater than that in class $j$. When the lower bound of $m_i$ is greater than the upper bound of $m_j$, it necessarily follows that $m_i > m_j$. Thus, when
> $$
> \frac{N_i - \epsilon_i}{S_{\max}} > \frac{N_j}{\rho S_{\min}},
> $$
> we can conclude a sufficient condition for $m_i > m_j$:
> $$
> \rho > \frac{N_j S_{\max}}{(N_i - \epsilon_i) S_{\min}}.
> $$
>
> To facilitate analysis, we assume $S_{\text{max}} \approx S_{\text{min}}$, meaning that the sizes of all mini-clusters are approximately equal. Then we obtain
> $$
> \rho > \frac{N_j}{N_i - \epsilon_i}.
> $$
> Therefore, if $\rho$ is too small or $\epsilon_i$ is too large, it may lead to the failure of $m_i > m_j$ even when $N_i > N_j$.
>
> Moreover, both the minimum purity $\rho$ of mini-clusters and the number $\epsilon_i$ of class $i$ samples located outside the mini-clusters dominated by class $i$ depend on the quality of the embeddings. If the embedding quality is low and the model struggles to consistently group samples of the same class into the same mini-clusters, the above property may fail to hold. Therefore, in MiniClustering, we use a model pre-trained with unsupervised representation learning to ensure a reasonable embedding distribution.
>
> We have already empirically demonstrated in Section 3.1 and Appendix H of the paper that head classes, which have higher true class frequencies, occupy more mini-clusters, revealing a clear relationship between mini-cluster assignment and true class frequencies. Now we have also provided a theoretical derivation showing that this phenomenon holds under favorable embedding distributions.
>
> We sincerely thank you for your careful review. Your insightful comments have significantly enhanced the depth of our work, and we greatly appreciate your valuable feedback.

---

### Author Response · Authors · 2025-11-25
**Looking forward to your further assessment**

Dear **Reviewers**,

Thank you for taking the time to review our manuscript and for your valuable feedback and recognition. We have carefully addressed all the comments and concerns raised, as reflected in our detailed responses and the revised manuscript and supplementary material.

We sincerely appreciate your efforts and look forward to your further assessment.

Best regards,

The Authors

---

### Author Response · Authors · 2025-12-03
**Global Response and Revisions Summary**

Dear Area Chair,

We sincerely thank you for your time and effort in reviewing our work. Here, we summarize it for quick overview.

First, **all reviewers appreciate this work**:

-	The method is **innovative** (KT3s, dtmG) and **practical** (QMja), offering **a new approach** to unsupervised re-weighting in clustering (amJw).
-	The experiments across different datasets and imbalance ratios are **comprehensive and supportive** (KT3s, QMja), **demonstrating reliable performance** even in scenarios with extreme imbalances and large-scale data (amJw). Detailed ablation studies provide **insight** how these components influence the clustering quality (KT3s, dtmG).
-	It provides variants **suitable for large-scale datasets** (KT3s) and **adaptable to various frameworks** (dtmG).
-	The paper is well-written (KT3s).

During the discussion, **we addressed all the concerns raised by each reviewer and made corresponding revisions to the manuscript** (highlighted in blue). Below is a summary of our responses to each reviewer.

We first summarize the issues raised by more than one reviewer.

**Common Concerns**:

-	We provide an empirical analysis of the relationship between true class frequencies and mini-cluster assignments for Reviewers KT3s and QMja. We find that classes with more samples dominate more mini-clusters, consistent with the core assumption of this paper. The relevant results are presented in Appendix H of the paper.
-	We have provided both Reviewers amJw and dtmG with a more detailed hyperparameter analysis and guidelines for choosing hyperparameters in the unsupervised setting, enhancing the practical utility of our method. The updated results are included in Appendix D of the paper.
-	We conduct experiments on text datasets using a Transformer-based BERT variant for Reviewer amJw and dtmG, demonstrating that our method remains superior performance (added in Appendix K) and also confirm our core assumption holds in different data types and encoder architectures.

Below, we summarize the concerns raised by only one reviewer.

**For Reviewer KT3s**:

-	We provide a detailed explanation of the measures we adopted to mitigate the impact of noisy pseudo-labels in early training and present experimental evidence demonstrating their effectiveness.
-	We explain and experimentally demonstrate how the mini-cluster strategy separates tail classes from head classes.
-	We ran multi-seed experiments to confirm our method's statistically significant performance gain. The results are added in Appendix I of the paper.
-	We use UMAP to visualize the learned embeddings in Appendix J of the paper. It can be observed that after training, tail-class embeddings become more distinguishable from head-class ones.
-	We have adjusted the manuscript layout to make Figures 2 and 3 easier to read.
-	We provide a theoretical foundation, deriving the inequality conditions required for our method to hold. It indicates our method requires high-quality mini-cluster partitions. We ensure this by using embeddings from a pre-trained representation learning approach in proposed MiniClustering. The detailed explanation and derivations can be found in Section 3.1 and Appendix M of the paper.

**For Reviewer amJw**:

-	We conducted experiments with various batch sizes, showing that our method consistently outperforms the self-labeling baseline across different batch sizes. Notably, our approach remains effective even when the batch size is smaller than the number of classes.

**For Reviewer QMja**:

-	We clarify that although some existing methods also employ over-clustering as an auxiliary clustering head, they have not been applied to long-tailed deep clustering. More importantly, those approaches neither explore the relationship between mini-cluster assignments and true class frequencies nor establish a connection between the target and auxiliary clustering heads for collaborative optimization.
-	We incorporate the reviewer's suggestion and revise the titles of Phenomenon 1 and Phenomenon 3 in Section 3.1.
-	We provide a more detailed explanation of the role of Phenomenon 2 to address the reviewer' confusion.
-	We conducted experiments on balanced datasets, demonstrating that our method still maintains strong performance in such settings. The results have been added to Appendix L of the paper.
-	We have discussed and cited the relevant methods provided by the authors in the paper.

**For Reviewer dtmG**:

-	We provide a detailed explanation of the MaxiClustering mechanism and have revised Appendix F to offer readers a more detailed description.
-	We conduct experiments by further training the baseline for 200 additional epochs and compared it with our method, providing a more comprehensive validation of our approach's effectiveness.

The above analysis and experimental results are detailed in our response, with relevant updates included in the revised manuscript. Thank you again for your time and effort.

Best regards,

The Authors

---

### Meta-Review · Area_Chair_mRtb · 2026-01-07

**Summary:**

The concerns raised by the reviewers have been largely addressed by additional experimental results and clarifications. Concerns regarding the theoretical motivation have been approached but are not deeply addressed. However, this is not required for a paper which can be seen as mainly empirical.

**Reviewer Concerns:**

Reviewer KT3s
* Lack of theoretical justification connecting mini-cluster counts to class long-tailedness; heuristic weighting without probabilistic/geometric reasoning
  * The authors provided a theorem with formal conditions, and derivations linking mini-cluster occupancy to class size under purity and representation quality assumptions.
  * The reviewer would likely still find the explanation insufficient because the necessary assumptions are parts of the reviewer's criticism; clearer clearer conditions, dependence on representation quality/geometry, and failure cases would be informative.
  * Thus is concern is only partially addressed
* Early-training pseudo-label noise across both heads; risk of unstable/bias weight updates; lack of mitigation
  * The authors clarified pretraining of the encoder, K-means centroid initialization, confidence thresholding; provided initial pseudo-label quality metrics showing strong filtered performance before training (after pre-training)
  * Hence this concern is addressed
* Sensitivity to mini-clustering quality/errors
  * The authors discussed the dependency on embedding quality and proposed using high-quality pretrained features; provided analyses of how mini-clusters separate tail classes and theoretical conditions.
  * This concern is partially addressed (empirical and theoretical discussion provided, but the reviewer still concerned about rigor).
* Statistical significance of improvements
  * Addressed.

Reviewer amJw
* Hyperparameter sensitivity and lack of automated adaptation
  * The authors provided an extended sensitivity analysis and practical guidelines.
  * This mainly addressed the concern but no automated/unsupervised tuning mechanism.
* Lack of non-image modality validation
  * Addressed by adding text experiments and showing improvements over baselines.
* Rules for designing number of mini-clusters and impact on stability
  * The authors addressed this mainly by making recommendations and explained performance trade-offs when M is too small or too large.
* Effect of batch size
  * Addressed by reporting results on Tiny ImageNet across batch sizes 64 to 512, consistently outperforming a self-labeling baseline.

Reviewer QMja
* Novelty relative to prior over-clustering (e.g., IIC, PICA); lack of direct connection to long-tailed clustering and head–tail estimation
  * The authors clarified differences to existing works but the degree of novelty can be considered limited.
* Phenomenon 3 not directly estimating class probabilities
  * The authors provided empirical correlations to substantiate their claims but the theoretical depth of this can be questioned.

Reviewer dtmG
* Hyperparameter sensitivity; lack of unsupervised tuning strategy
   * See above, mainly addressed.Authors: Provided practical guidelines and sensitivity study (Appendix D).
Status: Partially addressed. No automated tuning; guidance improves usability.
* MaxiClustering variant appears to contradict MiniClustering’s core logic
  * Clarified
* Implicit assumption of known number of classes K
   * Acknowledged by the authors; still an open issue but common also for other SoTA algorithms.
* Robustness of core assumption (Phenomenon 3) across data types/encoders
  * Addressed by additional results.

**Reviewer Scores:**

* Reviewer KT3s would not change their score from 4
* Reviewer amJw might increase their score to 6
* Reviewer QMja might increase their score to 6
* Reviewer dtmG would likely not change their score from 6

Overall the scores thus would have suggested a borderline paper.

---

### Decision · Program_Chairs · 2026-01-26

Accept (Poster)